# The RyR1 P3528S Substitution Alters Mouse Skeletal Muscle Contractile Properties and RyR1 Ion Channel Gating

**DOI:** 10.3390/ijms25010434

**Published:** 2023-12-28

**Authors:** Chris G. Thekkedam, Travis L. Dutka, Chris Van der Poel, Gaetan Burgio, Angela F. Dulhunty

**Affiliations:** 1Eccles Institute of Neuroscience, John Curtin School of Medical Research, Australian National University, Acton, ACT 2601, Australia; c.thekkedam@victorchang.edu.au; 2Department of Animal, Plant and Soil Sciences, School of Agriculture, Biomedicine and Environment (SABE), La Trobe University, Melbourne, VIC 3086, Australia; t.dutka@latrobe.edu.au; 3Department of Microbiology, Anatomy, Physiology and Pharmacology, School of Agriculture, Biomedicine and Environment, La Trobe University, Melbourne, VIC 3086, Australia; c.vanderpoel@latrobe.edu.au; 4Division of Genome Sciences and Cancer, John Curtin School of Medical Research, Australian National University, Acton, ACT 2601, Australia; gaetan.burgio@anu.edu.au

**Keywords:** ryanodine receptor, RyR1 P3528S substitution, mouse model, muscle contractile properties, Ca^2+^ activation of skinned muscle fibres, fibre-type composition, single ryanodine receptor activity

## Abstract

The recessive Ryanodine Receptor Type 1 (RyR1) P3527S mutation causes mild muscle weakness in patients and increased resting cytoplasmic [Ca^2+^] in transformed lymphoblastoid cells. In the present study, we explored the cellular/molecular effects of this mutation in a mouse model of the mutation (RyR1 P3528S). The results were obtained from 73 wild type (WT/WT), 82 heterozygous (WT/MUT) and 66 homozygous (MUT/MUT) mice with different numbers of observations in individual data sets depending on the experimental protocol. The results showed that WT/MUT and MUT/MUT mouse strength was less than that of WT/WT mice, but there was no difference between genotypes in appearance, weight, mobility or longevity. The force frequency response of extensor digitorum longus (EDL) and soleus (SOL) muscles from WT/MUT and MUT/MUT mice was shifter to higher frequencies. The specific force of EDL muscles was reduced and Ca^2+^ activation of skinned fibres shifted to a lower [Ca^2+^], with an increase in type I fibres in EDL muscles and in mixed type I/II fibres in SOL muscles. The relative activity of RyR1 channels exposed to 1 µM cytoplasmic Ca^2+^ was greater in WT/MUT and MUT/MUT mice than in WT/WT mice. We suggest the altered RyR1 activity due to the P2328S substitution could increase resting [Ca^2+^] in muscle fibres, leading to changes in fibre type and contractile properties.

## 1. Introduction

The ryanodine receptor (RyR) ion channel embedded in the membrane of the internal sarcoplasmic reticulum (SR) Ca^2+^ store in striated muscle is an essential component of excitation–contraction coupling. The ion channel releases the Ca^2+^ required for contraction in response to a signal from the dihydropyridine receptor (DHPR) in the surface/T-tubule membrane following membrane depolarisation. There are three isoforms of the RyR: RyR1 is the main isoform in skeletal muscle; RyR2 is the main isoform in cardiac muscle; and RyR3 is found in a variety of tissues. Amino acid substitutions in RyR1 are responsible for myopathies causing skeletal muscle weakness. The resulting disease phenotype is highly variable, sometimes with differences between affected individuals of the same family. The amino acid substitutions can be inherited as autosomal dominant or autosomal recessive mutations or they can occur spontaneously [1,2]. Dominant mutations generally compromise channel function, with many increasing the susceptibility to malignant hyperthermia (MH), although other mutations cause milder phenotypes with histology showing central cores as in Central Core Disease [3,4,5]. Recessive mutations including Multi-Minicore Disease, Centronuclear Myopathy or Congenital Fibre-type Disproportion are often associated with childhood onset and display a variety of histological abnormalities [5,6,7]. Greater disease severity is generally observed with recessive substitutions; however, there are exceptions such as the recessive RyR1 P3527S mutation that was identified in members of a consanguineous Algerian family, who suffered mild muscle weakness, but there was no family history of MH [8]. In that report, three of the affected patients demonstrated muscle weakness as infants with joint and hand involvement, as well as multiple minicores and hypertrophy of type I fibres in both upper and lower limb muscles. Biopsies from the same patients as adults demonstrated typical Central Core Disease histology with central cores containing rods in type I and type II fibres and hypertrophy of type I fibres. Multiple minicores were not present in the adult fibres and the myopathy was described as Central Core Disease with transient juvenile Multi-Minicore Disease. Studies using transformed lymphoblastoid cells from the affected patients showed that the mutation was associated with an increased resting cytoplasmic [Ca^2+^], while Ca^2+^ release after RyR1 activation by 4-chloro-m-cresol or caffeine was less than that in cells from unaffected patients [9]. 

While revealing dysfunction in Ca^2+^ handling, the transformed lymphoblastoid cells lack the normal post-translational modifications that define mature muscle fibre function including differentiation into distinct fibre types. These modifications could have major consequences for mature muscle fibre contractility, Ca^2+^ homeostasis and RyR1 ion channel activity. Thus far, there have been no reports of an animal model of the *human* P3527S mutation. The aims of the present study were firstly to create a *mouse* model of the mutation (i.e., *RYR1* P3528S) and then to explore the effects of the mutation on cellular and molecular aspects of muscle function. Our hypothesis was that the mutation in RyR1 would alter the channel activity and thus lead to changes in Ca^2+^ homeostasis in the muscle fibre, consequently perturbing muscle function. The mutation in mice was of additional interest because the majority of *RYR1* myopathies so far examined in animal models explore the effects of autosomal dominant *RYR1* mutations. Recessive mutations are less commonly examined [5]. Therefore, the *RYR1*-P3528S mouse provides an opportunity to explore the cellular effects of a recessive mutation. CRISPR-Cas9 gene editing was used to generate the P3528S mouse model. The results suggest that the mutation produces mild changes in the gating of RyR1 channels that could lead to an increase in the resting cytoplasmic Ca^2+^ concentration, which then affects the distribution of fibre types in specific muscles leading to contractile disfunction. 

## 2. Results

### 2.1. Mouse Appearance and Weight

There was no obvious effect of the P3528S mutation on overall mouse characteristics. The male and female MUT/MUT mice in all age groups were indistinguishable in appearance from WT/WT and WT/MUT littermates (Figure 1A). The differences between males and females were expected, as were the increases in weight with age. There was no significant difference between the weights of the three genotypes within the same sex and age group (Table 1). 

**Table 1 ijms-25-00434-t001:** Average weights of male and female mice belonging to each of the genotypes for young mice (1.8–6 months) and older mice (7–20 months). Data are given as mean ± sem with the number of mice in brackets (n). *****: old males significantly heavier than young males. *****: old females significantly heavier than young females. **^@^**: young females significantly lighter than young males. **^@^**: old females significantly lighter than old males.

	Young Males Weight (g)	Young Females Weight (g)	Old Males Weight (g)	Old Females Weight (g)
WT/WT	24.3 ± 0.8	20.5 ± 0.9 **^@^**	33.0 ± 1.7 *****	28.9 ± 1.2 *******^@^**
	(N = 8)	(N = 8)	(N = 5)	(N = 7)
WT/MUT	25.0 ± 0.9	18.5 ± 0.7 **^@^**	33.7 ± 1.0 *****	30.6 ± 1.0 *******^@^**
	(N = 6)	(N = 5)	(N = 15)	(N = 13)
MUT/MUT	26.7 ± 1.3	21.4 ± 0.8 **^@^**	33.7 ± 1.6 *****	27.6 ± 1.4 *******^@^**
	(N = 3)	(N = 6)	(N = 7)	(N = 10)

**Figure 1 ijms-25-00434-f001:**
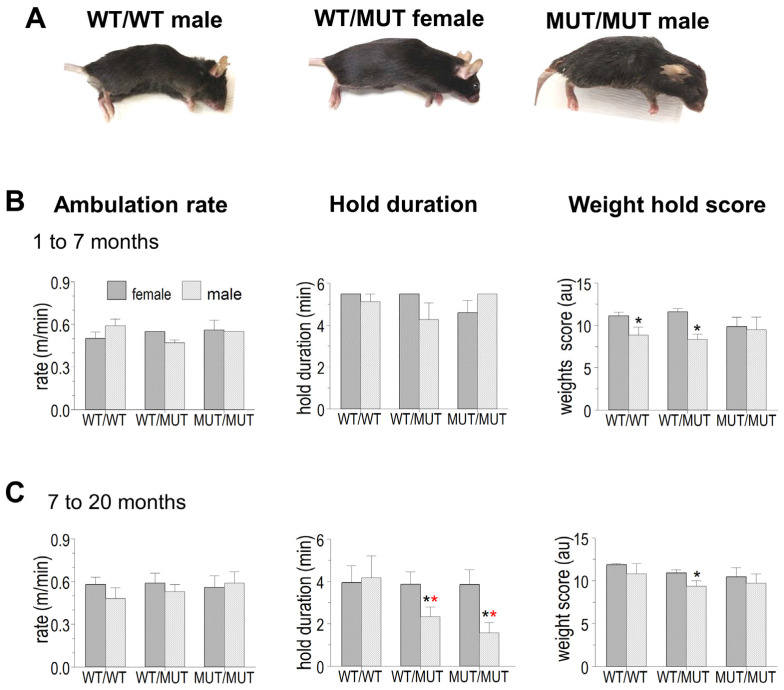
Effects of the RYR1 P3528S mutation on the appearance, ambulation rate and strength of WT/MUT and MUT/MUT mice compared to their WT/WT littermates. (**A**) Representative photographs of young mice taken immediately after CO_2_ euthanasia. From left to right: a WT/WT male, a WT/MUT female and a MUT/MUT male. (**B**,**C**) Average ambulation rate (m/min, left column), length of time that mice held onto the grid bar before letting go (hold duration (min) centre column) and the ability of mice to lift chains of increasing weight for 3 s or less, calculated as a weight score (detailed in the Section 4) (in arbitrary units (AU), right column). Data are separated into age groups with mice aged between 1.5 and 6 months in (**A**) and between 7 and 20 months in (**B**). The data were obtained from the same mice as those listed in Table 1. The number of mice in each group are given in Table 1. The data are shown as mean ± sem. The black asterisks indicate significant differences from between males and females. The red asterisks indicate a significant difference from WT/WT.

### 2.2. Mouse Mobility and Strength

The effect of the P3528S mutation on three aspects of live mouse function was assessed as outlined in the Section 4. These were (1) the rate of free ambulation; (2) the hold duration, which is the length of time (up to a maximum time of 5.5 min) that mice can hold onto a grid bar before voluntarily letting go; and (3) weight score that was determined from the number of chains of increasing weight that a mouse could hold for up to 3 s before voluntarily dropping the chain. The data were grouped by genotype and age (young (1.5–7 months) and older (7–20 months)). The results from the male and female mice with each of the three genotypes and in each of the two age groups were within similar ranges (Figure 1B,C). 

There were no consistent differences in the ambulation rates between males and females or between genotypes and there were no age-related differences (left hand graphs). The strength tests, however, did reveal significant differences between genotypes. The time that older male mice held onto the grid bar (hold duration, middle graphs) before releasing their grip was significantly less in WT/MUT mice compared to WT/WT and there was a further significant reduction in MUT/MUT mice. There was a trend towards a reduced hold duration in older female mice, but the difference was not significant. The weight score (right hand graphs) achieved by young male WT/WT mice was significantly lower than that achieved by young female WT/WT mice and there was a trend toward lower scores in both male and female older WT/MUT and MUT/MUT mice. Overall, the results indicated a small but significant loss of strength in the WT/MUT and MUT/MUT animals that was likely associated with muscle weakness. However, these tests alone did not preclude a loss of motivation from affecting the results in WT/MUT and MUT/MUT animals, particularly with the hold duration for the older MUT/MUT males. 

### 2.3. Contractile Properties of Isolated EDL and SOL Muscles

The 1 Hz twitch force (Pt), time to peak tension (TPT) and ½ relaxation time (½ RT) in either EDL or SOL muscles did not significantly differ between WT/WT, WT/MUT and MUT/MUT mice (Table 2). TPT is indicative of the myosin heavy chain composition and Ca^2+^ sensitivity of the contractile apparatus [10,11,12], whereas ½ RT is reflective of muscle relaxation and Ca^2+^ reuptake and depends on SERCA isoform and function [13]. The rate of force development (Dp/Dt) in response to a 1 Hz stimulation was also not significantly different in either EDL or SOL muscles between WT/WT, WT/MUT and MUT/MUT mice. 

To determine whether the genotype affects muscle force output in response to increasing frequency (1–120 Hz), force responses were expressed as a % of maximum force production (Fmax). Both EDL (Figure 2A) and SOL (Figure 2B) muscles from MUT/MUT mice presented with a downward shift in the force frequency curves when compared to both WT/WT and WT/MUT mice. At 10, 30, 50 and 60 Hz, the force produced by EDL muscles from MUT/MUT mice was significantly lower than those from both WT/WT and WT/MUT EDL muscles (*p* < 0.05, Figure 2A). In SOL muscles from MUT/MUT mice, the forces produced in response to 10 and 30 Hz stimulations were significantly lower than those from both WT/WT and WT/MUT mice (*p* < 0.05, Figure 2B). At frequencies above 60 Hz for the EDL, and above 30 Hz for the SOL, there were no differences in force production. When normalized to muscle cross-sectional area, the maximum specific force (sFmax) of WT/MUT and MUT/MUT EDL muscles was significantly lower than that of WT/WT EDL muscles (*p* < 0.05, Figure 2C). There was no statistical difference between the sFmax of soleus muscles (Figure 2D) from WT/WT, WT/MUT or MUT/MUT mice.

### 2.4. Mechanically Skinned Fibres

The properties of the contractile apparatus in each mechanically skinned fibre segment were determined by directly activating the contractile apparatus with heavily-buffered Ca^2+^ solutions with progressively higher free [Ca^2+^] (see Section 4.3.3). As shown in Figure 3 and Figure 4, the fibre type was functionally determined in real time by exposing the fibre to Sr^2+^-containing solution at pSr 5.3. In this way, the predominant isoform of TnC (either fast or slow) could be determined [14,15,16]. The subsequent “staircases” were repeated twice to determine reproducibility, and the plateau relative forces were then plotted to produce a force–pCa curve (Figure 3(Aiii,Bii) and Figure 4(Aii,Bii)) for each fibre and then the Hill coefficient “*h*” and pCa_50_ (an indicator of Ca^2+^-sensitivity) values calculated.

The number of type I, type II and mixed fibres examined and their relative proportions within EDL and SOL muscles across the three genotypes (WT/WT, WT/MUT and MUT/MUT) are summarized in Table 3. In the SOL muscle, there was a significant loss of pure type I and II fibres and an increase in the number of mixed fibres when compared to WT/WT to MUT/MUT mice, while the EDL was relatively unchanged (far right column, Table 3). Interestingly, the EDL muscle of MUT/MUT mice contained significantly more type I fibres than the WT/WT EDL muscle (Table 3).

The properties of the contractile apparatus (Ca^2+^-sensitivity “pCa_50_” and the Hill coefficient “*h*”) are summarized in Table 4. There was a significant difference in the Ca^2+^ sensitivity of approximately +0.08 pCa units in both the EDL and SOL muscles between WT/WT and MUT/MUT mice. There was no significant change in *h*.

### 2.5. RyR1 Single Channel Activity

The effect of the RyR1 P3528S mutation on the open probability and gating of the RyR1 ion channel was next examined using single channels incorporated into artificial lipid bilayers. RyR1 was obtained from mice in three age groups: young (3–4 months), middle-aged (10–11 months) and old (18–20 months). Channel activity was first measured with a cytoplasmic (*cis*) [Ca^2+^] of 1.0 µM. Then, the [Ca^2+^] was lowered to 300 nM, into the resting range, before adding caffeine at increasing concentrations from 10 µM to 10 mM. 

#### 2.5.1. Effects of the P3528S Mutation on Channel Characteristics

The conductance of channels from WT/MUT and MUT/MUT mice were not consistently different from those of WT/WT mice and the patterns of activity were similar (see records in Figure 5 and Figure 6). However, small differences between the three genotypes in open probability (*P_o_*) and gating parameters indicated the P3528S mutation had a mild effect on RyR1 channel gating. In particular, channels from old MUT/MUT mice were more sensitive to the change in *cis* [Ca^2+^] from 1 µM to 300 nM than channels from WT/WT or WT/MUT mice, as apparent in the recordings from individual channels from all genotypes and age groups with a decrease in activity when the [Ca^2+^] was reduced (Figure 5A–C). Although the average *P_o_* tended to decline in all cases, it was significantly lower only in channels from old MUT/MUT mice (inserts in Figure 5F), associated with a significant reduction in the frequency of opening (*F_o_*) in old MUT/MUT channels (Appendix A). Increasing the [caffeine] to 10 mM then produced a robust significant increase in *P_o_* in all genotypes and all ages, except for channels from old WT/MUT mice (Figure 5D–F). The activation was due to small significant increases in mean open time (*T_o_*) in most cases (Appendix A) and a substantial significant abbreviation of the closed times) (*T_c_*) in all genotypes and all ages (Appendix A), combining to produce the significant increase in *F_o_* (Appendix A) and *P_o_* (Figure 5). An increase in sensitivity to caffeine in old MUT/MUT channels was indicated by the significant increases in *P_o_* and *F_o_* induced by 10 µM caffeine, although the maximum *P_o_* with 10 mM caffeine was not consistently different from that of WT/WT in the WT/MUT or MUT/MUT channels (Figure 5 and Appendix A).

Changes in channel gating parameters with any experimental manipulation are often expressed as relative parameter values rather than absolute parameter values because of the usual wide range of parameter values in individual RyR channels (with *P_o_* varying from 0.0001 to 0.2, as shown in this study and in [17,18]). With this wide range of individual values, the average values reflect the results from the most active channels. Normalizing the data for each channel to its internal control value yields individual values and average data that are independent of intrinsic channel activity [18,19,20,21]. The results for parameter values expressed relative to values with 300 nM cytoplasmic Ca^2+^ are shown in Figure 6C–E and Appendix A and summarized Table 5. The largest significant change in relative *P_o_* in old MUT/MUT channels was due mainly to a significant increase in relative event frequency, thus mirroring the changes in the absolute parameters described above in this section. There was a consistent trend in all data sets towards an increase in relative *P_o_* and relative *F_o_* in 1 µM Ca^2+^, which indicates that the threshold for the Ca^2+^_-_activated increase in RyR activity in the absence of other channel modulators lies between 300 nM and 1 µM cytoplasmic [Ca^2+^]. It is notable that there was a trend for the relative *P_o_* and *F_o_* in WT/MUT and MUT/MUT channels to be greater than in WT/WT channels within each age group. If the relative *P_o_* values for all age groups are combined, the average relative *P_o_* for WT/MUT channels (2.08 ± 0.040, n = 33) was significantly greater (*p* = 0.040) than for WT/WT channels (1.37 ± 0.12, n = 38) and the average relative *Po* for MUT/MUT channels (2.02 ± 0.40, n = 32) was also significantly greater (*p* = 0.489) than that of WT/WT channels. The significance increased to *p* = 0.02 for the difference between the relative *P_o_* for WT/WT channels and the combined average relative *P_o_* for all channels carrying the P2328S mutation (WT/MUT plus MUT/MUT channels (2.02 ± 0.27, n = 65)). The significant increase in *P_o_* in the old MUT/MUT channels alone, and the significant increase in the combined data suggests that the threshold for activation of the channels in muscle fibres from the WT/MUT and MUT/MUT animals could be reduced and may contribute to an increase in cytoplasmic [Ca^2+^].

The relative changes in channel activity resulting from caffeine addition shown in Figure 6 and Appendix A are summarized in Table 6. As with the parameter values described above (Figure 5), the greatest increase in average relative *P_o_* was seen in all cases with 10 mM caffeine and was attributed a small increase in open time and a large reduction in closed durations with significant increases in relative *F_o_* in all genotypes (Appendix A and Table 6). The response of the channels to increasing caffeine concentrations was also generally similar in the three genotypes with the greatest increase in relative *P_o_* with a caffeine concentration of 10 mM, and with smaller but substantial significant increases in relative *P_o_* with 1 mM caffeine in channels from oldest mice of the three genotypes (Figure 6). An increased sensitivity to caffeine is suggested by the large significant increase in relative *P_o_*, with 10 µM caffeine in old MUT/MUT channels (Table 6). Together these changes in absolute and relative parameters suggests that the P3528S substitution may increase the sensitivity of the channel to 10 µM caffeine and also an increase in the efficacy of 10 mM caffeine in channels from the oldest animals. 

#### 2.5.2. An Age-Dependent Increase in Activity

An unexpected observation in the WT/WT data was the age-dependent increase in channel activity that is apparent in the recordings in Figure 5A–C and in the average *P_o_* data (Figure 5D–F). The average *P_o_* of channels from old WT/WT mice was significantly greater than in channels from young WT/WT mice with both 1 µM and 300 nM Ca^2+^ (Figure 5D–F). The higher *P_o_* with 1 µM and 300 nM Ca^2+^ was due to a significantly higher event frequency (*F_o_*) (Appendix A) with 300 nM Ca^2+^ due to significantly shorter closed durations (*T_c_*) in channels from middle-aged mice (Appendix A). There were no significant age-dependent differences in channel open durations (*T_o_*) (Appendix A). Because the increase in activity was similar at both Ca^2+^ concentrations, there was no change in the average relative *P_o_* data (Table 5 and Figure 6). Age-dependent effects were less apparent in WT/MUT and MUT/MUT channels. 

### 2.6. Histological Assessment

The effect of the P3528S mutation on the cross-sectional profile of fibres was assessed for the EDL and SOL muscles from young (2–4 months) and older (6–12 months) WT/WT, WT/MUT and MUT/MUT mice. Representative areas of transverse sections of the EDL and SOL muscles from the 2- to 4-month-old mice are shown in Figure 7A,B. There was no indication of lesions or of central cores in WT/MUT or MUT/MUT muscles from either age group. The sections appear similar to those in a previous study of EDL and SOL muscles from wild type *C57BL6J* mice [22]. The EDL sections contain fibres in two distinct size groups (Figure 7A), which we classified as group A and group B. The larger WT/WT EDL group A fibres had an average cross-sectional area of 808.9 ± 54.4 µm^2^ and the WT/WT EDL group B fibres had a significantly smaller average cross-sectional area of 270.5 ± 28.9 µm^2^ (Table 7). The average cross-sectional area of EDL group A fibres decreased significantly in muscles from WT/MUT and MUT/MUT mice but remained significantly larger than group B (Table 7). 

The cross-sectional areas of WT/WT SOL fibres showed little variability with ~98% of fibres (SOL group A) having a relatively large diameter (645 ± 24 µm^2^). The SOL group A fibres were significantly smaller than the EDL group A fibres in all genotypes. There was no significant difference between WT/WT, WT/MUT and MUT/MUT in the cross-sectional areas of the SOL group A fibres (Table 7). The smaller SOL group B were seen only occasionally in younger mice (Table 7). 

The average percentage of larger EDL group A fibres from young mice was significantly lower in WT/MUT and MUT/MUT fibres than in WT/WT fibres (Figure 7C). The average percentage of group A fibres in EDL muscles from the older mice was similar in WT/WT and WT/MUT fibres but fell significantly in older MUT/MUT fibres (Figure 7D). There were corresponding significant increases in the relative numbers of smaller EDL group B WT/MUT and MUT/MUT fibres (Figure 7C,D). In contrast, in the soleus, there were no significant differences between the genotypes in the relative numbers of group A or group B fibres from mice aged 2–3 months. However, the percentage of SOL group B fibres was significantly greater in mice aged 6–12 months than in younger animals of all genotypes (Figure 7D), with a significant increase in SOL group B fibres from MUT/MUT mice. 

The fibre type distribution in wild type *C57BL6J* mice has been determined using histochemical and biochemical techniques [22]. The result showed that the larger EDL fibres are type IIB and that the smaller EDL fibres are a mixture of type IIX and IIX/B hybrid fibres, while the larger group A fibres in SOL are predominantly type I and type IIA/X hybrid and type IIX fibres. In that study, IIX/B hybrids occurred in 20% of the EDL fibres and IIA/X hybrids in 19% of the SOL fibres. The smaller SOL group B fibres that we occasionally observed may correspond the type IIA/X hybrid fibres which can assume a wide range cross-sectional areas. It is likely that most of the SOL fibres in our WT/WT *C57BL* mice were type IIX because we detected very little type I MyHC in mouse hindlimb or forelimb muscles (Figure 8 below). We have assigned the commonly detected fibre types identified by Augusto et al. [22] to the EDL and SOL groups in Figure 7 for the sake of discussion, with the caveat that a further histochemical and a more rigorous MyHC study would be required to confirm any apparent changes in fibre type resulting from the P3528S mutation.

### 2.7. Myosin Heavy Chain (MyHC) Composition

The MyHC composition of the hindlimb muscle from WT/WT, WT/MUT and MUT/MUT mice and from rat EDL and SOL muscles was examined in muscle homogenates (Figure 8). The four isoforms are clustered closely between the 150 kDa and 250 kDa. Rat EDL and SOL muscle MyHC isoforms are well characterized [13,22] and were used here as controls for identifying the MyHC bands in our mouse muscle homogenates. The band corresponding to type IIB was further confirmed in a preliminary Western Blots analysis (see Appendix A). 

Figure 8A shows expanded lanes of a gel comparing control homogenate bands from rat EDL and SOL muscles (right lanes) with mouse hindlimb muscles (left lane). The rat SOL muscles mainly contained the MyHC type I, while the rat EDL muscles contained type IIA, IIB and IIX, as expected [13,22]. The mouse hindlimb muscle homogenates in the left lane contained mainly Type IIB and IIX, with a faint band at the MyHC type I molecular mass level. The MyHC composition in mouse lower limb muscles supports the interpretation of the histology data for the EDL, where the majority of fibres were thought to be Type IIB and a smaller percentage were type IIX or type IIX/B. The very low density of the almost undetectable MyHC type I band and relatively low density of the type IIX band in the hindlimb muscles from the mice used in our study suggests that mouse SOL muscles are mainly composed of type IIX fibres. 

There were no consistent differences between the relative MyHC densities in muscles from WT/WT, WT/MUT and MUT/MUT mice, or in the relative amounts of the myosin IIB and IIX in muscles from young or older animals or between homogenates from hindlimb and forelimb muscles (Figure 8). A similar apparent lack of difference between the band densities for the three genotypes was seen in another three silver-stained gels (see Appendix A). These results suggest that the fibre type changes that we see in the EDL and SOL muscles in the histology and skinned fibre experiments are specific for these muscle groups. It is likely that the relatively small changes seen in the histological sections (Figure 7) were beyond the resolution of SDS-PAGE of combined limb muscles, where the volume of EDL and SOL muscles is relatively small. 

## 3. Discussion

Overall, many effects of the RyR1 P3528S substitution in mice are consistent with the phenotype described in humans and in their transformed lymphoblastoid cells [8,9]. In all cases, the effects are relatively mild. The strength of the WT/MUT and MUT/MUT mice was less than that of the WT/WT mice, without changes in physical characteristics or longevity. There were small significant changes in the contractile properties of MUT/MUT EDL and SOL muscles, with less force generated at lower frequencies and a reduction maximum twitch force in fast-twitch EDL muscles. The pCa_50_ for Ca^2+^ activation of mechanically skinned fibres was shifted by approximately 255 nM to a lower [Ca^2+^] in MUT/MUT EDL and SOL fibres, with an increase in the percentage of type I fibres in the EDL and increase in the percentage of mixed type I and type II fibres in the SOL. Both muscles appeared normal in histological sections without obvious structural damage to the muscles or to individual fibres. However, the percentage of small-diameter fibres increased in EDL muscles from WT/MUT and MUT/MUT mice and there was a similar trend in the SOL. It might be expected that the immediate effect of the mutation would be on RyR1 structure and function and we observed changes in the ion channel gating in the presence of cytoplasmic [Ca^2+^] near resting levels that could lead to elevated resting cytoplasmic [Ca^2+^] in the muscle fibres. 

### 3.1. Elevated Cytoplasmic [Ca^2+^] Could Alter Expression of Factors Determining Fibre Type

It would not be surprising if a small increase in the resting cytoplasmic [Ca^2+^] had a feedback effect on the expression or translation of some or all of the myriad of factors that determine the fibre type composition of specific muscles and their contractile characteristics [13]. Fast-to-slow transitions have been reported in EDL fibres with an increase in resting cytoplasmic [Ca^2+^] from 100 nM to 500 nM [23] and in cultured myotubes when the intracellular [Ca^2+^] was increased 3–10 fold [24]; both results suggest a relationship between [Ca^2+^] and fibre type expression. However, the pathways linking cytoplasmic [Ca^2+^] to gene expression are complex and likely to involve Ca^2+^-binding proteins such as calmodulin which, in itself, has numerous functions in muscle fibres, including the regulation of RyR1 activity by binding to the protein in close proximity to residue 3527/8. Fibre type transformation is also a complex process involving changes in all muscle cell compartments, including components of the excitation–contraction coupling, cell metabolism and contractile machinery [13]. However, although unlikely, we cannot exclude the possibility that the altered response of WT/MUT and MUT/MUT RyR1 channels to changes in cytoplasmic Ca^2+^ is a secondary effect on the ion channel resulting from fibre type transformation [25]. This then begs the question of which process initiates the change in fibre type? Indeed, the answer could lie outside of the muscle fibre because there is also an important non-contractile embryonic role of RyR1 and/or RYR1-mediated Ca^2+^ signalling during muscle organ development [26,27]. Finally, there is the possibility that the RyR1 mutation has neuronal effects that would impact muscle activation [26], as discussed below (Section 3.4).

### 3.2. Potential Consequences of Increased RyR1 Activity with 1 µM Cytoplasmic Ca^2+^ on Ca^2+^ Homeostasis

We suggest that the small but significant increase in relative RyR1 open probability with 1 µM *cis* Ca^2+^ compared to 300 nM *cis* Ca^2+^ with the P3528S mutation could lead to an increase RyR1 leak and therefore contribute to the approximately 10–30 nM increase in cytoplasmic [Ca^2+^] found in immortalized lymphocytes from patients both heterozygous and homozygous for the P3528S mutation [9]. Our data similarly showed significant effects of the mutation on ion channels from WT/MUT and MUT/MUT mice. Despite the increase in cytoplasmic [Ca^2+^], Ducreux et al. [9] concluded that the mutation did not increase RyR1 leak because the size of the SR Ca^2+^ store (i.e., the Thapsigargin-sensitive Ca^2+^ pool) was unaltered in the MUT/MUT cells. However, it is possible for an increase in leak to occur without a change in the size of the store if an appropriate leak/pump balance is maintained as in cardiac myocytes [28] and perhaps with compensation from surface membrane Ca^2+^ entry mechanisms such as store-operated Ca^2+^ entry (SOCE). Additionally, a depletion in Ca^2+^ may occur in the micro-domains of the SR and terminal cisternae in close proximity to RyR1 release sites, but not contribute significantly to the total store Ca^2+^ content. It is notable that there was a significant decline, or strong trend towards a decline, in specific twitch force in EDL muscles from WT/MUT and MUT/MUT mice and SOL muscles from MUT/MUT animals, which could reflect a small depletion of Ca^2+^ in the micro-domains of the terminal cisternae’s Ca^2+^ store.

The effect of RyR1 activators on Ca^2+^ release can provide another indicator of changes in cellular Ca^2+^ handling. This was attempted with the transformed lymphocytes; however, measurements of the cellular responses to 4-chloro-m-cresol (4-cmc) and caffeine are not straight forward, with several variables including the numbers of cells responding to the activators [9]. The conclusion in that study was that the P3527S substitution reduced the amount of Ca^2+^ released in response to activation by 4-cmc and caffeine but that the P3527S mutation did not significantly reduce the sensitivity of RyR1 to activation by either caffeine or 4-cmc. In contrast, we found a significant increase in sensitivity to caffeine in RyR1 channels from old MUT/MUT mice. 

### 3.3. Fibre Type Changes Indicated in Mechanically Skinned Fibres and in Fibre Cross-Sectional Areas

The results of the histological examination of EDL and SOL muscles were consistent with the results from application of the Sr^2+^-containing solution to mechanically skinned fibres. Both techniques suggest that there are changes in the fibre populations in EDL and SOL muscles. As discussed above, the skinned fibre experiments showed that EDL muscles from MUT/MUT mice contained significantly more type I fibres than EDL muscles from WT/WT mice and there was a strong trend towards an increase in the fraction of type B fibres in EDL muscles from WT/MUT mice (Table 3). Although the changes in cross-sectional area (Figure 7) indicate that the P3528S substitution has an effect on the fibre population, size does not define fibre type because the cross-sectional areas of specific fibre types can vary between different muscle groups and different fibre types can have similar cross-sectional areas [22]. Nevertheless, the increased fraction of small fibres in EDL muscles from MUT/MUT mice is consistent with the appearance of type I fibres in those muscles. Skinned SOL fibre responses to Sr^2+^-containing solutions indicated a significant reduction in the percentage of pure type I and type II fibres with an increase in the percentage of mixed fibres in WT/MUT and MUT/MUT mice compared with WT/WT mice (Table 3). The change from pure type I and type II fibres to mixed fibres may not necessarily change the cross-sectional areas of SOL fibres as both fibre types have a similar cross-sectional area [22] and both would contribute to the larger diameter SOL fibres.

### 3.4. The P3538S Substitution Could Alter Mouse Behaviour

We cannot discount the possibility that the physical weakness that we observed in WT/MUT and MUT/MUT mice was due to the mice lacking motivation to continue with the tasks where the end-point involved mice voluntarily dropping the “load”. It is well established that RyR1, RyR2 and RyR3 are expressed in neurons [29]. Changes in the activity of the RyRs can alter neuronal function. For example, RyR2 is located in Purkinje cells of the cerebral cortex and excessively active “leaky” RyR2 channels associated with mutations causing cardiomyopathies can result in seizures [30]. Substantially reduced voltage-dependent Ca^2+^ release in hypothalamic nerve terminals of mice heterozygous for the human loss-of-function I4898T Central Core Disease mutation [26]. Mutations in RyR1 accessory proteins can also alter RyR1 activity with neuronal consequences. One example is the chloride intracellular channel (CLIC2) protein expressed in skeletal muscle, cardiac muscle and the brain. Its best-defined physiological role is inhibiting RyR1 and RyR2 channel activity [31]. The substitution reverses the inhibitory action of CLIC2 on RyR1 and RyR2, so that the H101Q CLIC2 becomes a strong activator of the ion channels and is associated with seizures, cognitive impairment and learning disabilities [31]. Thus, the excessive release of Ca^2+^ from internal stores can trigger excess neurotransmitter release and affect other post-synaptic pathways including those involved in neuromuscular function. 

### 3.5. Possible Effects of the P3528S Substitution on RyR1 Structure and Ion Channel Gating

The RyR1 residue 3528 is located towards the C-terminal end of Helical Domain 2 (3016–3572, also known as a part of the clamp domain and α-solenoid). It is located near the junction with the Central Domain (3668–4251) [32]. Importantly, the Central Domain is the only cytoplasmic domain that is continuous with the transmembrane domain and ion channel gating residues that control channel activity. The greatest mobility in RyRs is in the clamp domain which undergoes a downward rotation upon channel opening and contains binding sites for ligands including the FK506-binding proteins calmodulin and CLIC2 [21,32,33]. As a result of the functionally important mobility of this region, its high resolution structure is not well defined, with parts of the backbone structure remaining undefined [32]. The proline to serine substitution may have profound effects on the local structure in this region, as also suggested for the P2328S substitution in RyR2 [28], because proline residues have a strong influence on protein structure and can introduce functionally important directional changes into a helix. Substitution with an uncharged serine residue could reverse the influence of the proline and disrupt the structure and function of the surrounding region of the native protein. The pathways for the transmission of local structural effects of the mutation from peripheral sites to the channel pore in the distant transmembrane domain may well follow a defined route through the central and C-terminal domains, as has been elegantly shown for the malignant hyperthermia mutation [34,35].

### 3.6. The P3528S Substitution on One Allele Is Sufficient to Alter Function

It is notable the results of the very different parameters, measured in three different laboratories, show trends and significant changes in WT/MUT characteristics in the same direction as the trends and significant changes in MUT/MUT properties, albeit often to a lesser degree in the WT/MUT context than in MUT/MUT animals. An impact of the substitution on WT/MUT characteristics was seen in the mouse strength, in the muscle force frequency, in Ca^2+^ sensitivity and fibre type determined in mechanically skinned fibres, in fibre cross-sectional areas and in the functional properties of the RyR1 ion channel. These observations indicate that the P3528S substitution on one allele may indeed be sufficient to alter muscle characteristics. In the homozygous situation, four subunits of the RyR would carry the P3528S substitution, whereas in the heterozygous situation, there would be a distribution between channels containing one, two, three or four subunits carrying the mutation. In this case, the average effect of the mutation on the overall protein structure and function would depend on (a) the percentage of channels carrying one, two, three or four mutated subunits and (b) on the location of the mutated residue, its influence on the immediately surrounding structures and on inter-subunit interactions. It is conceivable that the mutation in one subunit alone could distort the central portions of the tetrameric cytoplasmic assembly and its influence is transmitted through the Central Domain to the channel pore and gating residues. 

### 3.7. Embryonic Importance of RyR1

It is worth noting that we used the same CRISPR-Cas9 gene editing approach to generate mice with known autosomal dominant RyR1 myopathy-associated mutations including ***R403C*** [36], ***R614L*** [37], ***R4214-F4216*** *del* [38] and ***L4578V*** [39]. These attempts were not successful, possibly due to the essential nature of RyR1 in embryonic development indicated by homozygous RyR1 null mice dying shortly after birth with small limbs and abnormal skeletal muscle structures [27]. Microarray analyses revealed that 318 genes are differentially expressed in homozygous RyR1 null embryos and that the expression of multiple transcription factors are altered as well as components of key signalling pathways [27]. The human CCD substitution in RyR1 I4898T expressed in mice (RyR1 I4899T) is neonatally lethal in homozygous animals and heterozygous mice have significantly impaired mobility with reduced voltage-activated Ca^2+^ transients in hypothalamic magnocellular terminals [26]. Residue I4898 is located in the RyR1 channel pore and the equivalent mouse I4899T substitution results in reduced current flow through the pore [26]. The generation of viable P3528S mice was likely due to the mild phenotype of this substitution. 

### 3.8. Possible Limiting Factors

Not all studies were performed on the same animals. The muscle contraction and skinned fibre studies were performed on a single shipment of 15 old littermates (5 of each genotype) sent to La Trobe University. The remainder of the experiments were conducted on animals at the ANU before and following euthanasia, with muscles from one side of the mouse fixed for histology and muscles from the contralateral side snap frozen and later processed separately either for SR/RyR1 channel isolation or for myosin examination. Tissues from more than one animal were often combined to obtain sufficient material for the experiments. 

In addition, the number of animals (N) used in the different experiments and the number of observations “n” reported varied with the different experiments. For the animal studies and isolated muscle experiments, “n” experiments were equal to the number of animals (N). For skinned fibres, “n” equalled the number of fibres examined. Since several fibres were obtained from one muscle from one mouse, “n” is greater than the number of animals (Table 3 and Table 4). For the single channel data, “n” is the number of observations, which is double the number of experiments because independent data were obtained at both positive and negative potentials, with current flowing through the pore in opposite directions at each potential. Furthermore, muscles from one to three mice of the same genotype and age were processed to obtain SR vesicles enriched in RyR1. Finally, two to three separate preparations were often required to obtain sufficient numbers of channel recordings because of the nature of the technique where channel incorporation into bilayers is notoriously variable as is the survival time of the bilayer following channel incorporation, with some bilayers lasting only tens of seconds. Here, the experiments lasted more than 30 min with seven solution changes so that the number of completed experiments was substantially less than the number of channel incorporations. The smaller number of experiments in some cases, particularly for older mice, could have influenced the results from these data sets. However, confidence was provided by trends in the average data being in the same direction as the significant changes and, as mentioned in Section 2.5.1, when the data from all age groups were combined to increase the overall number of observations, the differences in relative *P_o_* between WT/WT and WT/MUT or MUT/MUT mice were significant. 

However, an investigation of the myosin isoform distribution in specific muscle groups is required to further investigate the genotype-dependent changes in fibre type in the EDL and soleus muscles that we reported in the skinned fibre (Section 2.4) and histology experiments (Section 2.6). The data provided by the preliminary myosin isoform analysis (Figure 8) of total forelimb and hindlimb muscles were included because it is informative and the lack of difference between genotypes is a significant observation. 

## 4. Material and Methods

### 4.1. Generation and Genotyping of the RyR1-P3 Mice

The RYR1-P3528S mouse was generated by the ANU Transgenesis Facility at the JCSMR using CRISPR-Cas9 gene editing as previously described [40]. Briefly, the P3528S mutation was introduced into the *RYR1* gene by microinjection of the Cas9 nuclease protein (50 ng/µL), complexed with its single guide RNA (50 ng/µL) and a single-stranded oligonucleotide donor (50 ng/µL) into the pronucleus of *C57BL/6Ncrl*-derived fertilized eggs. All constructs were ordered from Integrative DNA Technologies (IDT). Correctly targeted founder mice were genotyped by PCR and Sanger sequencing to confirm the heterozygous substitution.

The single guide RNA (sgRNA) sequence was 5′-CATGTGTGCT**CCC**ACCGACC **AGG-3′**. 

The long single stranded (ssOligo) donor repair template sequence for the P3528S mutation was **5′-**CTCCGTGCAGACATCCTTGATCGTGGCCACACTCAAGAAGATGTTGCCGATCGGACTAAACATGTGTGCT**TCC**ACCGACC**AGG**ACCTCATTGTGCTGGCCAAAGCCCGCTATGCCCTGGTGCTTATCCAGCCCCACCCAGT-**3′**.

The P3528S substitution was achieved with the C to T nucleotide substitution as indicated.

The genotype was determined with PCR and Sanger sequencing. The forward primer *RYR1*P3528S_SF was 5′-ATAGGTTGTGGTTGGTGCTG-3′ and the reverse primer *RYR1*P3528S_SR was 5′-GACGTTCTGGAGAGCTTTGG-3′. 

WT/MUT males and females were selected as breeders for experimental animals. The litters contained expected ratios of WT/WT, WT/MUT and MUT/MUT pups. Breeding was continued for 8 generations. Animals of ages from 2 to 20 months were used in various experiments and the data were collected into age groups as described the Results section and Figure legends.

The P5328 mouse has been cryopreserved [41] and the mice are available from Jackson Laboratories, Bar Harbor, Maine, United States, as JAX Stock #033136.

*Ethics approval for mouse generation in the ANU Transgenesis Facility* was given by the ANU Animal Experimentation Ethics Committee (AEEC) under the protocol #A2014/07 in agreement with the National Health and Medical Research Council (NHMRC) Australian code for the use of animals for scientific purpose.

### 4.2. Animal Experimentation

#### Preliminary Phenotype Assessment

***Ambulation***. The ambulation rate of mice during free exploration of a 52 × 31 cm container was assessed. The animal was placed in a cage and allowed free movement. The distance that the mouse covered was measured from a video recorded over a 3 min period following a ~5 min settling period [42]. Ambulation rate is expressed as m/min.***Strength tests***. Two strength tests were used [43,44]. In the first test, the mice were placed onto a wire screen that was then inverted and held ~30 cm above a soft surface for a maximum period of 5.5 min. The mice clung to the screen with their forelimbs before either dropping onto the soft surface or being lowered onto the surface if they maintained their grip for 5.5 min. The time that the mice held onto the screen (“hold duration”) was recorded. In many cases, mice held onto the screen for the 5.5 min period with the “hold duration” achieving a “glass ceiling” of 5.5 min. In the second test, the ability of mice to hold weights constructed of chains with increasing numbers of rings, for period of up to 3 s, was examined. The minimum weight was 18 g (2 rings) and the heaviest was 45 g (5 rings). Each set of rings was attached to fine wire tangles that the mouse grabbed with its front paws when held above the tangles. The mouse was presented with the lightest chain first; if that chain was held for 3 s, the animal was rested for ~1 min and then presented with next heaviest chain and so on up to the heaviest chain. Generally, mice were able to hold between 27 and 36 g for 3 s without dropping the chain. A “hold score” was calculated based on the chain weight and the length of time that the weight was held. Weight #1 (2 rings) was assigned a value of **1**; weight #2 (3 rings) a value of **2**; weight #3 (4 rings) a value of **3**; and weight #4 (5 rings) a value of **4**. Therefore, for example, if weights #1 and #2 were held for 3 s and weight #3 for 1 s, the score in arbitrary units (AU) was calculated as (**1** × 3) + (**2** × 3) + (**3** × 1), giving a score of 12 AU.

### 4.3. Euthanasia and Tissue Collection

Mice were euthanized by CO_2_ asphyxiation following completion of the ambulation and strength tests and then weighed and muscle tissues were immediately removed. Muscle tissues for SDS-PAGE, Western blotting, myosin heavy chain (MyHC) assessment and RyR1 channel recording were isolated from limbs on one side of the mouse and immediately snap frozen in liquid N_2_ and stored at −80 °C. Muscles for histological examination were collected from limbs on the contralateral side of the mouse and immersed in a formalin solution. Two adult male Sprague Dewey rats used as controls for MyHC analysis (Section 4.5 below) were similarly euthanized, and the EDL and SOL muscles were immediately removed, snap frozen in liquid N_2_ and stored at −80 °C. Ethics approval for breeding, euthanasia and tissue removal was included in ANU AEEC protocols #A2017/40 and #A2015/35.

#### 4.3.1. Whole Muscle Experiments

On the day of experimentation, the mice were anaesthetized via intraperitoneal injection of sodium pentobarbitone (6 mg/kg), such that they were unresponsive to tactile stimuli. Isolated muscles from one side of the mouse were used to measure the isometric contractile properties of EDL and SOL muscles, as described in detail previously [45]. EDL and SOL muscles from the contralateral side were used for the skinned fibres experiments (see Section 4.3.2). Briefly, isolated muscles were tied at the proximal and distal tendons with braided surgical silk, surgically excised and transferred to a 1300A Whole Mouse Test System (Aurora Scientific, Aurora, ON, Canada) organ bath filled with Krebs Ringer solution containing (in mM) NaCl 137, NaHCO_3_ 24, D-glucose 11, KCl 5, CaCl_2_ 2, NaH_2_PO_4_H_2_O 1, MgSO_4_ 1, d-tubocurarine chloride 0.025, bubbled with Carbogen (5% CO_2_ in O_2_, BOC gases, Preston, Australia), and thermostatically maintained at 25 °C. The distal tendon of the isolated muscle was tied to an immobile pin and the proximal tendon was attached to the lever arm of a dual mode force transducer (300-CLR, Aurora Scientific, Aurora, ON, Canada). EDL and SOL muscles were stimulated via two platinum electrodes that flanked the length of the muscle using a supramaximal voltage of 0.2 ms square wave pulses of 350 and 800 ms train duration, respectively. All stimulation parameters and twitch and twitch contractile responses were controlled and measured using Dynamic Muscle Control Software (DMC v5.415), with on board controller interfaced with the transducer control/feedback hardware (Aurora Scientific, Aurora, ON, Canada).

Maximal twitch force at the optimal muscle length was determined by delivering a 1 Hz stimulation pulse every 30 s, with micromanipulations of muscle length in between pulses, until a plateau in the maximum peak twitch (Pt) force occurred. Following 4 min of rest, maximum isometric tetanic force (Fmax) production was determined from the plateau of the force frequency curve, whereby the EDL muscles were stimulated at 10, 30, 50, 60, 80, 100 and 120 Hz, and the SOL at 10, 20, 30, 50, 60, 80 and 100 Hz with a 2 min rest between stimulations. Muscle cross-sectional area was determined following the guidelines outlined in the standard operating procedure for measuring the isometric force of isolated mouse muscles ex vivo by TreatNMD (https://treat-nmd.org/ 9 September 2022) [46]. Absolute maximum tetanic force (Fmax) was normalized to muscle cross-sectional area and expressed as specific force (sFmax, kN/m^2^). 

#### 4.3.2. Single-Fibre Preparation for Skinned Fibre Experiments

Contralateral EDL or SOL muscles were pinned at resting length in a Petri dish lined with Sylgard 184 (Dow Corning, Midland, MI, USA) and immersed in paraffin oil and kept cool (~10 °C) on an icepack. Using jeweller’s forceps, segments of individual fibres (~25–30 µm diameter) were mechanically skinned (see Lamb 2002) and mounted at 120% of resting length on a force transducer (AME801, Horten (model AME-801; SensorOne, Sausalito, CA) with a resonance frequency of >2 kHz. The fibre segment was then transferred to a 2 mL Perspex bath containing standard K^+^-based solution that broadly mimics the myoplasmic environment (see below). 

#### 4.3.3. Solutions for Mechanically Skinned Fibre Experiments

All chemicals listed in 4.3 to 4.5 were purchased from Sigma-Aldrich (St. Louis, MO, USA), unless specified otherwise. Examination of the properties of the contractile apparatus required the use of heavily Ca^2+^-buffered solutions with either 50 mM EGTA (i.e., relaxing solution) or 50 mM CaEGTA (i.e., maximum Ca^2+^-activating solution), as described previously [47]. Additionally, a strontium-based solution (at pSr 5.3) was made by mixing relaxing solution (50 mM EGTA) with Sr-EGTA solution similar to the maximum Ca^2+^-activating solution; this solution was used to functionally determine the fibre type or the presence of “mixed” fibres (pSr 5.3 force level < 5% for type II and >65% for type I fibres [14,15,16,47]). These solutions contained (in mM) total ATP, 8; Na^+^, 37; K^+^, 126; free [Mg^2+^], 1; phosphocreatine, 10; and HEPES, 90 with the pH adjusted to 7.100 and an osmolarity of 295 ± 5 mOsm/kg. 

#### 4.3.4. Contractile Apparatus Measurements of Skinned Fibres

The contractile apparatus properties of mechanically skinned EDL and SOL muscle fibres were characterized by directly activating the contractile apparatus with a series of heavily Ca^2+^-buffered solutions of different known pCa values (where pCa = −log_10_ [Ca^2+^]). After a 2 min wash period in relaxing solution, the skinned fibre segment was exposed to a range of progressively stepwise higher free [Ca^2+^] in solution (0.1–20 μM). These solutions were produced by mixing appropriate amounts of the 50 mM EGTA and the 50 mM CaEGTA solutions. This sequence of solutions elicited force responses resembling a “staircase” (see Figure 3 and Figure 4). The isometric force responses elicited at each pCa value were expressed as a percentage of the corresponding maximum Ca^2+^-activated force (“max.”), plotted against the pCa value and a Hill curve fitted using GraphPad Prism 6 (GraphPad Software, San Diego, CA, USA). In the case of mixed fibres, which displayed a combination of both type I and type II contractile properties (functionally identified by sensitivity to pSr 5.3), force–pCa curves were proportionately and appropriately adjusted [16,48].

### 4.4. Single-Channel Electrophysiology

#### 4.4.1. Vesicle Preparation

SR vesicles containing RyR1 were prepared from mouse limb muscles using a mini-preparation procedure developed for mouse cardiac SR [18]. Protease inhibitors were added to pre-prepared homogenizing buffer (HB), containing (in mM) sucrose 290, imidazole 10, and NaN_3_ 3, pH 6.9, adjusted with 6 M HCl and Storage Buffer (SB) composed of HB + 649 mM KCl. Individual protease inhibitors were added at the following final concentrations: benzamidine hydrochloride hydrate, 1.0 mM; pepstatin A, 2.1 µM; leupeptin, 1 µM; AEBSF/Pefabloc SC, 0.5 mM; calpain inhibitor I, 3 µM; and calpain inhibitor II, 3 µM. All steps were performed on ice or at 4 °C. Muscle tissues from one mouse (~1 g) were homogenized in 2 mL of HB. The homogenate was then centrifuged at 12,000× *g* for 20 min, the pellet (P1) was discarded, and the supernatant (S1) was centrifuged at 43,000× *g* for 2 h. The P2 pellet was resuspended in SB, homogenized in a Potter homogenizer for 15 min, then centrifuged at 46,000× *g* for 90 min. The P3 pellet was resuspended in 160 µL of SB in a Potter homogenizer, then aliquoted into 12 µL tubes, snap frozen and stored at −80 °C.

#### 4.4.2. Lipid Bilayer Formation and Vesicle Incorporation

As described previously [20], lipid bilayers were formed from phosphatidylethanolamine, phosphatidylserine and phosphatidylcholine (5:3:2 *w/w*) (Avanti Polar Lipids, Alabaster, AL, USA) painted across an aperture with a 200 µm diameter in the wall of a 1.0 mL Delrin (Cadillac Plastics, Silverwater, NSW, Australia) cup manufactured in house. SR vesicles (10 µg/mL) were added to the *cis* chamber. The cytoplasmic side of channels incorporated into the bilayer faced the *cis* solution. Bilayer potential was controlled, and single channel currents were recorded using an Axopatch 200A amplifier (Axon Instruments, Foster City, CA, USA). Bilayer potential is expressed as V*cis* − V*trans* (i.e., Vcytoplasm − Vlumen). Bilayers were formed and vesicles were incorporated using a *cis* solution containing (in mM) 230 caesium methanesulfonate (CsMS); 20 CsCl; 1 CaCl_2_; and 10 tetraethylsulfamide (TES), pH 7.4 (adjusted with CsOH), and a *trans* solution containing (in mM) 30 CsMS, 20 CsCl, 1 CaCl_2_; and 10 TES (pH 7.4). Following ion channel incorporation, (a) the *cis* solution was replaced by perfusion with an identical solution, except that the [Ca^2+^] was 1.0 µM and (b) 200 mM CsCH3O3S was added to the *trans* chamber for symmetry. Channel activity was recorded continuously throughout the experiment. Current was filtered at 1 kHz with a lowpass 8-pole Bessel filter and sampled at 5 kHz using the in-house analogue/digital conversion program BLM. These parameters give a sampling interval of 200 µs or one point every 200 µs. Additional digital filtering allowed the channel to be visualized in real time using BLM. The bilayer potential was switched every 30 s between −40 mV, 0 and +40 mV. In the present experiments, control channel activity was first recorded with 1.0 µM *cis* Ca^2+^ before the concentration was dropped to 300 nM by the addition of BAPTA (at a concentration determined using a Ca^2+^ electrode (Radiometre Analytical ISE25Ca Ca^2+^ electrode (Villeurbanne, Cedex, France)) to better approximate the resting cytoplasmic [Ca^2+^]. Channel activity recorded with 300 nM *cis* Ca^2+^ was used as the control activity to normalize the data to determine the relative parameters for activity recorded with 1 µM *cis* Ca^2+^ and the following addition of caffeine.

#### 4.4.3. Analysis of RyR1 Channel Activity

Channel activity was analysed at +40 mV and at −40 mV over periods of approximately 90 s, i.e., 3 segments of continuous activity at each potential, under each condition. Slow fluctuations in the baseline were corrected using an in-house baseline correction program (written by Dr. D. R. Laver, University of Newcastle). Channel activity was measured using the in-house programs Channel 2 (developed by PW Gage and M Smith, JCSMR, Australian National University, ACT, Australia) or Channel 3 (developed by NW Laver, University of Newcastle, NSW, Australia), using a threshold analysis. Threshold levels for channel opening and closing were set to exclude baseline noise at 20% of the maximum single channel conductance. The analysis yielded values for open probability (*P_o_*) and for the gating parameters mean open time (*T_o_*), mean closed time (*T_c_*) and event frequency (*F_o_*). Open probability and gating parameters are presented as the measured parameter values and are also expressed relative to internal control values with 300 nM *cis* Ca^2+^ calculated for each individual channel. The relative changes in activity gave equal weighting to changes in activity of all RyR channels where control *P_o_* values varied from 0.0002 to 0.2 [17,18,21].

### 4.5. Histology and Myosin Heavy Chain Analysis

#### 4.5.1. Histology

Specific muscle groups were dissected for histology immediately after CO_2_ euthanasia and immersed in a solution containing 10% formalin phosphate-buffered saline (pH = 6.8) and then embedded (using standard Histology procedures in the JCSMR Flow Cytometry Facility). The muscle was sectioned transversely and stained with haematoxylin and eosin (HE). EDL and SOL muscles were examined in detail. Representative areas were photographed and cross-sectional areas of 30–80 abutting fibres in those areas were measured.

#### 4.5.2. Myosin Heavy Chain Analysis

**Tissue Preparation.** The procedure was modified from Talmadge and Roy (1993) [49]. Frozen limb muscle from two mice of a similar ages and genotype were combined for each preparation or 100 mg of frozen rat EDL and SOL muscles was used. The tissue was thawed in 2 mL Eppendorf tubes and covered in sufficient Myosin Heavy Chain Homogenizing Buffer (MyHC-HB) containing (in mM) 250 sucrose, 100 KCl, 5 EDTA, 40 DTT and 20 Tris-HCL, pH 6.8. Protease inhibitors were added containing (in mM) 1 benzamadine, 0.5 PMSF, 3000 anti-calpain I, 3000 anti-calpain II, 2300 leupeptin and 1.460 pepstatin A. The tissue was homogenized 3 times for 5 s each, and then centrifuged at 13,500× *g* for 20 min at 4 °C in an Eppendorf 5425R centrifuge. The pellet was resuspended and homogenized in 600–800 µL of MyHC-HB using a Potter homogenizer. The protein concentration of the pellet suspension was determined using the Bicinchoninic Acid Assay (BCA) and adjusted to 0.6 mg/mL for mouse muscles or 0.3 mg/mL for rat muscles and then stored at −80 °C.**MyHC SDS-PAGE**. The methods were based on Mizunoya et al. (2008) [50], with minor modifications to improve separation of proteins with molecular weights between 150 and 250 kDa. The running gel (8% *w*/*v* acrylamide-bis acrylamide (99:1), 35% *v*/*v* glycerol, 0.2 M Tris-HCl (ph-8.8), 0.1 M glycine, 0.4% *w/v* SDS, 0.1% *w/v* ammonium persulfate (APS), 0.05% *v/v* N,N,N′,N′-tetramethylethylenediamine (TEMED)) was polymerized for 1 h at room temperature. The stacking gel (4% *w/v* acrylamide-bis acrylamide (49:1), 30% *v/v* glycerol, 0.07 M Tris-HCl (ph-6.7), 4 mM EDTA, 0.4% *w/v* SDS, 0.1% *w/v* ammonium persulfate (APS), 0.05% *v/v* TEMED) was polymerized for 1 h at room temperature, and then overnight at 4° C. The wells were loaded with 20 µL of the muscle samples and run at a constant voltage of 70 V for 24 h in a cold room (~4° C) with constant stirring. The lower running buffer contained (in mM) 50 Tris, 75 glycine and 0.05% *w/v* SDS. The upper running buffer contained 300 mM Tris, 450 mM glycine, 0.3% *w/v* SDS and 0.12% *v/v* β-ME. After electrophoresis, the gels were either silver-stained (Bio-Rad, Cat. No. 1610449) or transferred onto polyvinylidene difluoride (PVDF) membranes for immunoblotting.**Western blots**. MyHC gels were equilibrated in transfer buffer (37 mM Tris, 140 mM glycine, 20% methanol) for 30 min and then transferred onto PVDF membranes using a wet transfer system under the following conditions: constant voltage of 100 V for 60 min followed by 30 min starting at 0.5A. Membranes were blocked in 5% skim milk in phosphate-buffered saline with 0.05% Tween 20 (PBST) for 1 h and subsequently incubated in the primary antibody overnight at 4° C. The membranes were then washed several times in PBST and incubated in the HRP-conjugated secondary antibody for 2 h at room temperature. They were then washed several times in PBST and imaged in the Bio-Rad ChemiDoc system after exposure to the ECL substrate (Thermo Fisher SuperSignal West Pico PLUS Chemiluminescent Substrate, Cat. No. 34577). Bands corresponding to type I, type IIB, type IIA and type IIX myosin isoforms were identified by their molecular masses relative to type IIB, which was confirmed by Western Blot, using the primary Myosin IIb Antibody#3404 (Cell Signaling Technology, Danvers, MA, USA). The secondary antibody was goat α-rabbit S-2004 obtained from Santa Cruz Biotechnology Inc., Santa Cruz, CA, USA.

### 4.6. Statistics

#### 4.6.1. Phenotype Characteristics, Lipid Bilayer Electrophysiology and Histology

Data in each of these sections are presented as mean ± sem and significance (*p* < 0.05) was evaluated using Student’s *t*-test or sign test. 

#### 4.6.2. Muscle Contractility

All values are presented as mean ± sem or sd where stated, with “n” denoting the number of individual fibres examined and “N” denoting the number of animals. When examining changes in fibre type proportions within EDL and SOL muscles of the three cohorts, statistical significance (*p* < 0.05) was determined using a Chi-square (χ^2^) goodness of fit test. One-way ANOVAs were used to analyse differences in contractile properties in EDL and SOL muscles between WT/WT, WT/MUT and MUT/MUT mice.

## 5. Conclusions

Using a mouse model of the *RYR1* P3528S mutation, we showed significant small but consistent effects of the P3528S substitution on the gating of the ion channel, on muscle contractile properties and on fibre types as well as on mouse strength. Significant changes were observed in all parameters in both heterozygous and homozygous genotypes, indicating that the presence of the substitution on only one or two of the four subunits of the RyR1 protein was sufficient to impact on the gating of the ion channel with flow-on effects on Ca^2+^ homeostasis and from there to a variety of other muscle characteristics. 

## Figures and Tables

**Figure 2 ijms-25-00434-f002:**
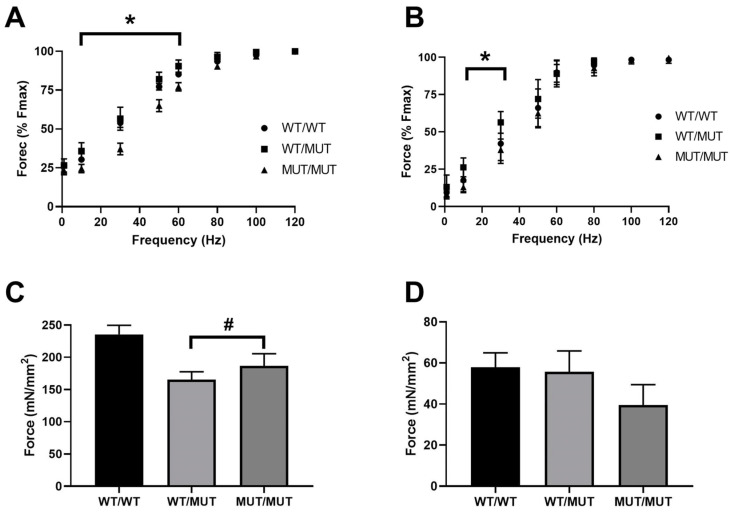
Effects of the RYR1 P3528S mutation on the contractile properties of EDL and SOL muscles from WT/MUT and MUT/MUT mice compared to their WT/WT littermates. The force response of EDL muscles (**A**) and SOL muscles (**B**), expressed as a % of Fmax in response to increasing stimulation frequency. Force output produced by EDL muscles (**C**) and SOL muscles (**D**), normalized to muscle cross-sectional area (specific force, sFmax). * *p* < 0.05—MUT/MUT mice compared their WT/WT littermates, as determined by two-way ANOVA; # *p* < 0.05—MUT/MUT and WT/MUT mice compared WT/WT mice, as determined by one-way ANOVA. WT/WT mice (N = 6), WT/MUT mice (N = 4) and MUT/MUT mice (N = 5).

**Figure 3 ijms-25-00434-f003:**
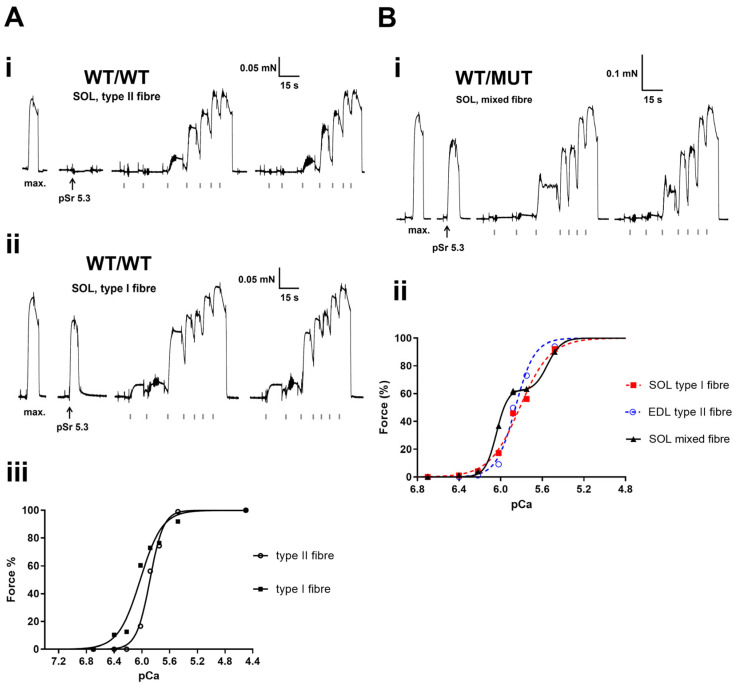
Typical example of a type II (**Ai**) and type I fibre (**Aii**) obtained from the same SOL muscle of a wild type mouse (WT/WT). Force responses elicited when directly activating the contractile apparatus with heavily Ca^2+^-buffered solutions with progressively higher free [Ca^2+^] (grey ticks under the black force trace indicate the pCa of successive solutions: >9.00, 6.70, 6.40, 6.22, 6.02, 5.88, 5.75, 5.48, 4.50 then back into >9.00 (repeated to test reproducibility)). A maximum Ca^2+^-activated force “max” response was elicited first before exposing the fibre segment to pSr 5.3. (**Bi**) “Mixed” fibre force recording of a SOL muscle obtained from a WT/MUT mouse. Note “force oscillations” at ~40% force in the first staircase and a plateauing of force around ~60% maximum Ca^2+^-activated force. Stimulus artefacts caused when transferring the fibre segment from one bath to the next, are truncated. (**Aiii**) Hill “*h*” fits to the force pCa staircases from Ai (pCa_50_ = 5.883 pCa units and *h* = 4.429) and (**Aii**) (pCa_50_ = 6.023 pCa units and *h* = 2.716). Similarly, (**Bii**) shows *h* fits for three consecutive fibres, one obtained from an EDL muscle and two from a SOL muscle from one WT/MUT mouse. The responses of these fibres verify that the changes in Ca^2+^ sensitivity are a manifestation of the physical properties of the fibre’s myosin content, not errors associated with the test solutions or other possible experimental variables. The pCa_50_ and *h* values in Bii are 5.901 pCa units and 2.625 (SOL type 1 fibre; red curve), 5.858 pCa units and 4.702 (EDL type II fibre; blue curve), and 5.816 pCa units and 2.958 (SOL mixed fibre, black curve), respectively. Note: Hill fits in (**Aiii**,**Bii**) are for the individual fibres not the average values.

**Figure 4 ijms-25-00434-f004:**
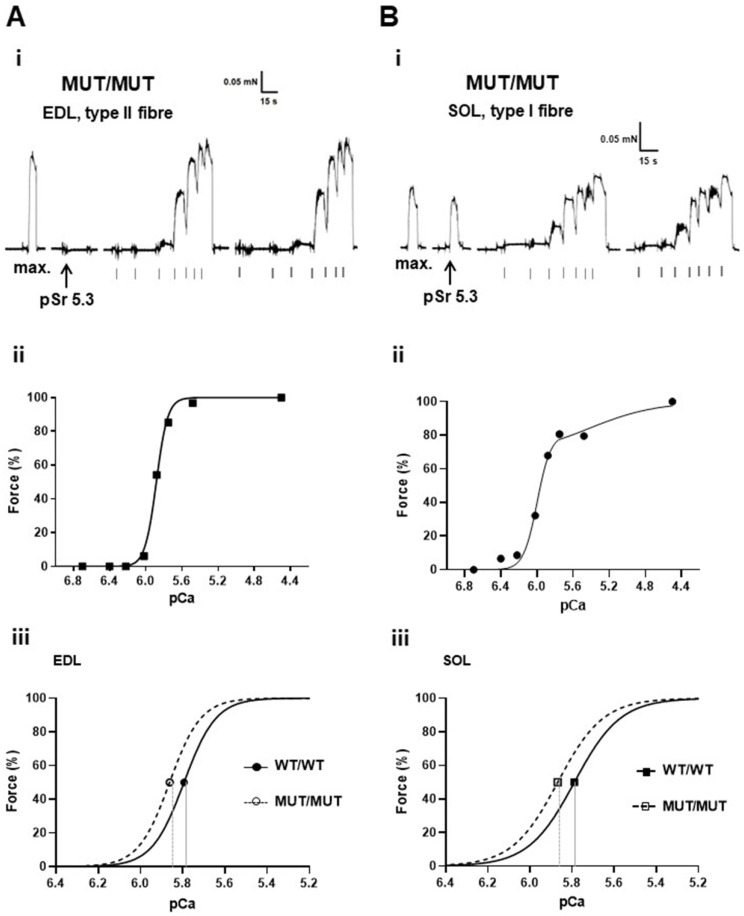
Typical examples of a type II fibre obtained from an EDL muscle in (**Ai**,**Aii**) and a type I fibre from a SOL muscle in (**Bi**,**Bii**) from one MUT/MUT mouse, with generalized responses shown in (**Aiii**,**Biii**). In both (**Ai**,**Bi**) panels, the maximum “max” force responses were elicited after exposing the fibre segment to pCa 4.5 and then the pSr 5 response was evoked. This was followed by two force “staircases” elicited by directly activating the contractile apparatus with heavily Ca^2+^-buffered solutions with progressively higher free [Ca^2+^] (grey ticks under the black force trace indicate the pCa of successive solutions: >9.00, 6.70, 6.40, 6.22, 6.02, 5.88, 5.75, 5.48, 4.50 then back into >9.00). Panels (**Aii**) and (**Bii**) are Hill “*h*” fits to the force pCa staircases from (**Ai**) and (**Bi**), respectively. The force traces in (**Ai**,**Bi**) in are from two fibres sequentially tested on the same day using the same solutions. The responses verify that the changes in Ca^2+^ sensitivity were a manifestation of the physical properties of the fibre’s myosin content, not errors associated with the test solutions or other possible experimental variables. Generalized shifts in Ca^2+^-sensitivity in EDL and SOL fibres from MUT/MUT mice are shown in panels (**Aiii**,**Biii**).

**Figure 5 ijms-25-00434-f005:**
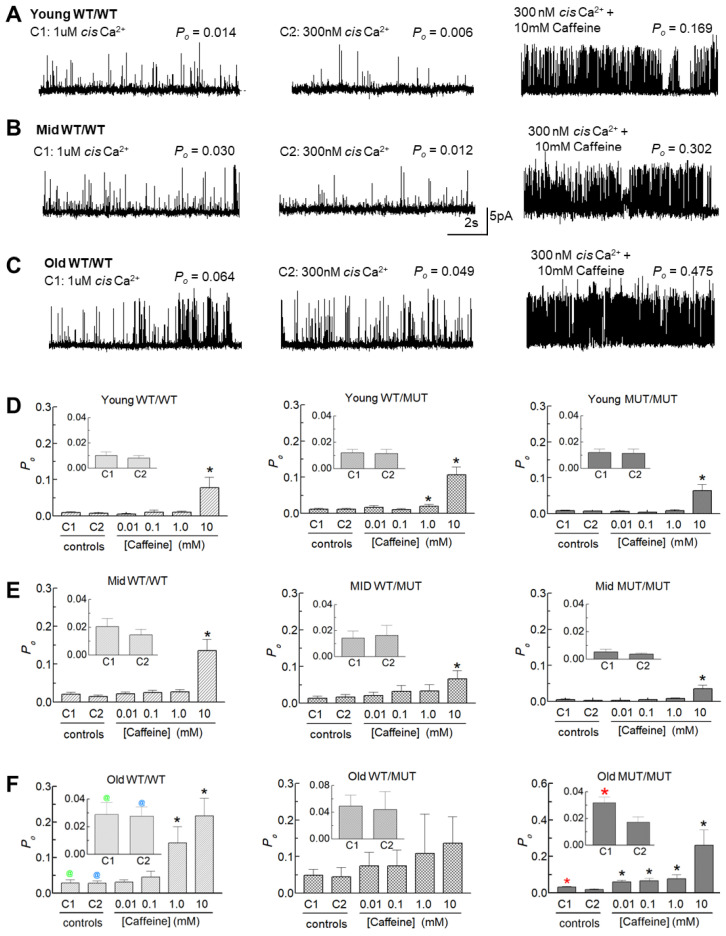
Single RyR1 channel activity in WT/WT channels increases with age and the effect of reducing Ca^2+^ from 1 µM to 300 nM is significant only in MUT/MUT channels. (**A**–**C**) Representative 10 s records from WT/WT RyR1 channels showing the age-dependent changes in activity recorded with cytoplasmic (cis) [Ca^2+^] of 1 µM (left), then 300 nM (middle) and then after increasing cytoplasmic [caffeine] to 10 mM while the cis [Ca^2+^] was maintained at 300 nM (right). Recordings are from young (**A**), middle-aged (**B**) and old (**C**) WT/WT mice. Channel activity is shown at +40 mV, with openings upward from the closed state at the bottom of each single channel record. The records in each row were obtained from the same RyR1 channel. *P_o_* values are given above each record. (**D**–**F**) Graphs of average channel open probability (*P_o_*) with 1 µM cis Ca^2+^ (C1), 300 nM cis Ca^2+^ (C2) and then following progressive increases in cytoplasmic [caffeine] to 10 µM, 100 µM, 1 mM and 10 mM. The inserts in each graph are expanded to show the effects of reducing cis [Ca^2+^] from 1 µM (C1) to 300 nM (C2). Average *P_o_* is shown for channels from WT/WT (left), WT/MUT (middle) and MUT/MUT (right) mice from the young (**D**), middle-aged (**E**) and old (**F**) groups. Data are shown as mean ± sem. The average values include *P_o_* at +40 mV and −40 mV. The number of observations (n) for each group are as follows: young WT/WT, n = 14; young WT/MUT n = 16; young MUT/MUT n = 12; middle-aged WT/WT n = 16; middle-aged WT/MUT n = 16; middle-aged MUT/MUT n = 14; old WT/WT n = 8; old WT/MUT n = 8; old MUT/MUT n = 6. ***** or *****, significantly different from C2 (300 nM cis Ca^2+^); **^@^**, significantly different from C1 (1 µM cis Ca^2+^) in channels from young WT/WT mice; **^@^**, significantly different from C2 (300 nM cis Ca^2+^) in channels from young WT/WT mice.

**Figure 6 ijms-25-00434-f006:**
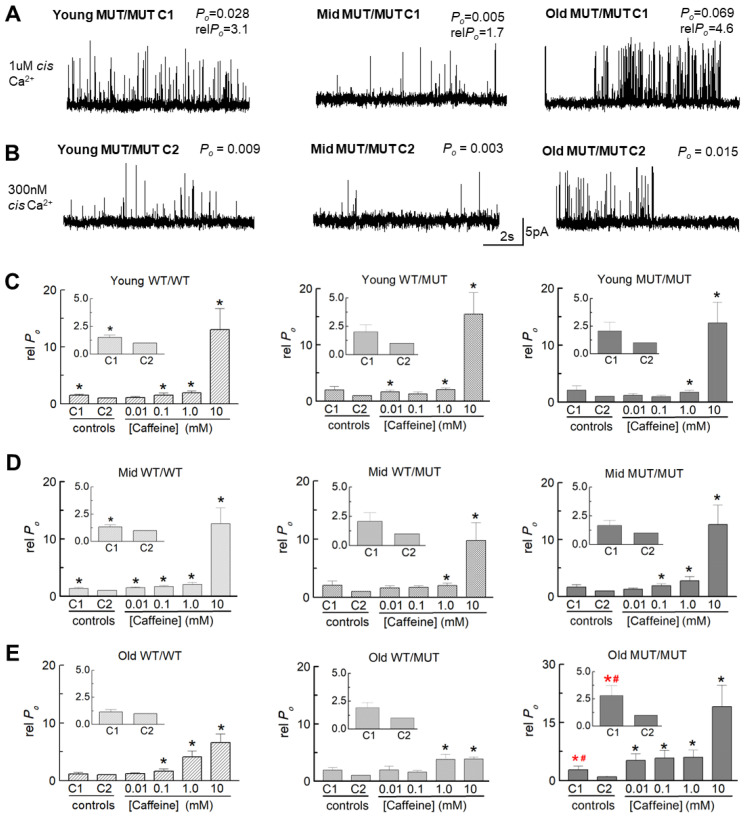
The activity of individual MUT/MUT RyR1 channels is higher with 1 µM cis Ca^2+^ than with 300 nM cis Ca^2+^ in most situations and the average relative *P_o_* is significantly greater in channels from old MUT/MUT mice than in channels from old WT/WT mice. (**A**,**B**) Representative 10 s records from MUT/MUT RyR1 channels from each age group showing reduced activity with a reduction in cytoplasmic [Ca^2+^] from 1 µM (**A**) to 300 nM (**B**). The channels in (**A**,**B**) were isolated from young (left), middle-aged (centre) and old (right) MUT/MUT mice. Channel activity is shown at +40 mV, with openings upward from the closed state at the bottom of each single channel record. The recordings in (**A**,**B**) in each age group are from the same channel. The *P_o_* for each record is shown above the record and the corresponding relative P_o_ values are given in (**A**). (**C**–**E**) Graphs of average open probability relative to C2 (300 nM cis Ca^2+^) (rel *P_o_*) are plotted for 1 µM cis Ca^2+^ (C1), and after progressive increases in cytoplasmic [caffeine] to 10 µM, 100 µM, 1 mM and 10 mM. The inserts in each graph are expanded to show the effects of reducing cis [Ca^2+^] from 1 µM (C1) to 300 nM (C2). Average rel *P_o_* is shown for channels from WT/WT (left), WT/MUT (middle) and MUT/MUT (right) mice, in young (**C**), middle-aged (**D**) and old (**E**) groups. The average values include rel *P_o_* at +40 mV and −40 mV. The number of observations (n) for each group are as follows: young WT/WT, n = 14; young WT/MUT n = 16; young MUT/MUT n = 12; middle-aged WT/WT n = 16; middle-aged WT/MUT n = 16; middle-aged MUT/MUT n = 14; old WT/WT n = 8; old WT/MUT n = 8; old MUT/MUT n = 6. ***** or *****, significantly different from C2 (300 nM cis Ca^2+^); ^#^, significantly different from rel *P_o_* of old WT/WT channels.

**Figure 7 ijms-25-00434-f007:**
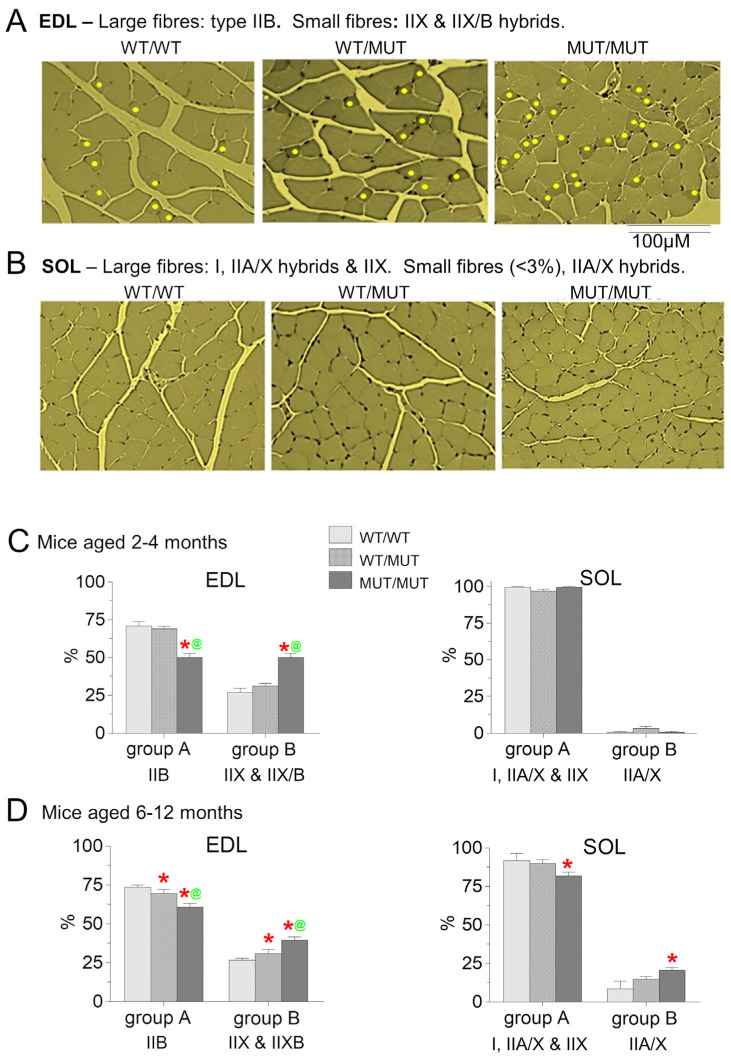
The cross-sectional profile of fibres differs in EDL muscles from WT/MUT and MUT/MUT mice compared with that of WT/WT mice. (**A**,**B**) show representative micrographs of EDL (**A**) and SOL (**B**) muscles from WT/WT mice (left), WT/MUT mice (centre) and MUT/MUT mice (right) aged between 2 and 4 months. The EDL muscles contained two distinct populations of fibres: larger EDL group A fibres and smaller EDL group B fibres. Smaller EDL group B fibres in (**A**) are marked with a yellow dot. The magnification is the same for panels (**A**,**B**). The scale bar below the right image in panel (**A**) applies to all images in (**A**,**B**). (**C**,**D**) show graphs of the average % of fibres in each group in EDL and SOL muscles from mice from each of the three genotypes aged 2–4 months (**C**) and 6–12 months (**D**). The cross-sectional areas of ~34–80 fibres were measured in each muscle and the fibres were allocated to group A or group B. The average % fibres in each group is shown as mean ± sem. ***,** significant difference between the % of fibres from WT/WT mice and WT/MUT or MUT/MUT mice. **^@^**, significant difference between the % of fibres from WT/MUT and MUT/MUT mice. Number of mice in the 2–4-month-old group (**C**): WT/WT, 17; WT/MUT, 14; MUT/MUT, 11. Number of mice for the 6–12-month-old group (**D**): WT/WT, 3; WT/MUT, 5; MUT/MUT, 11.

**Figure 8 ijms-25-00434-f008:**
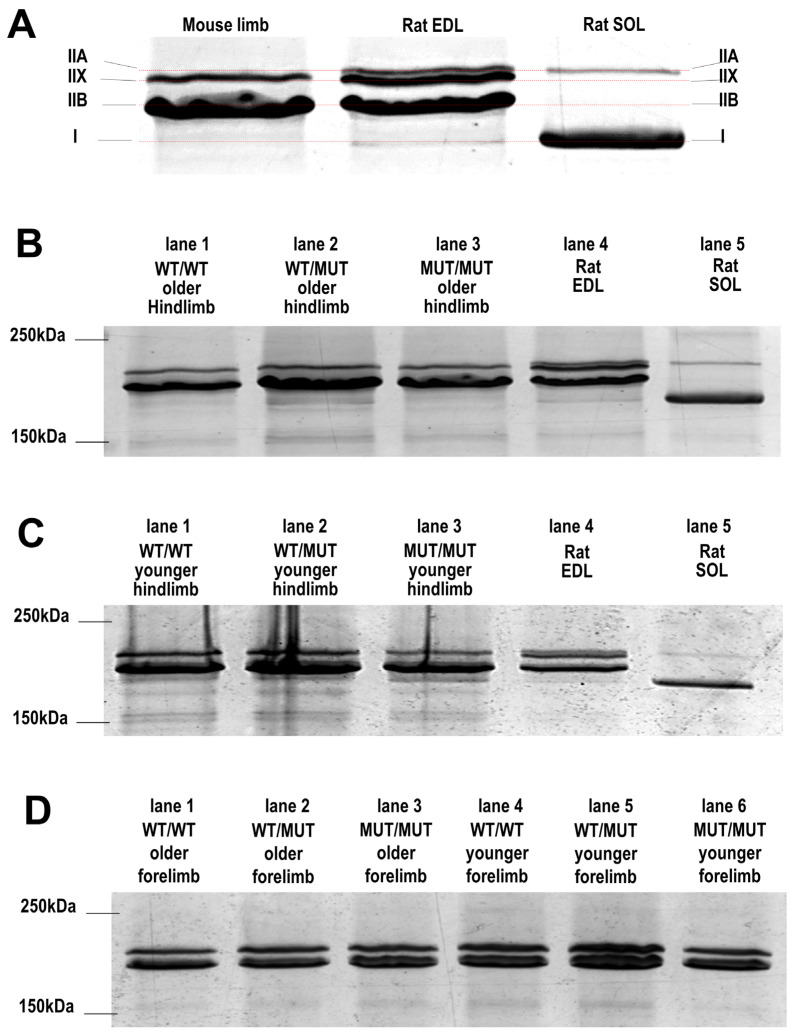
Silver-stained SDS-PAGE gels confirm fibre types assigned to individual fibres in histological sections of EDL and SOL muscles. (**A**–**D**) show bands between 150 kDa and 250 kDa in representative silver-stained SDS-PAGE gels of muscle homogenates. The image in (**A**) is an expansion of lanes 3, 4 and 5 lanes of the gel shown in (**B**) in order to compare bands in mouse homogenates (in this case MUT/MUT, lane 3) with those in rat EDL (lane 4) and rat SOL (lane 5) samples. The expansion illustrates the assignment of myosin isoforms to the bands in mouse homogenates, irrespective of genotype. The higher molecular mass IIA and IIX bands are well defined in rat EDL, while the IIA band is absent from rat SOL and from mouse limb muscles. (**B**–**D**) Comparison of myosin isoform in mice of the 3 genotypes and show the lack of a genotype dependence. (**B**). Hindlimb muscles from older (18–19 months) WT/WT, WT/MUT and MUT/MUT mice (lanes 1–3) and rat (~6 months) EDL and SOL muscles (lanes 4 and 5). (**C**)**.** Hindlimb muscles of younger (7–8 months) WT/WT, WT/MUT and MUT/MUT mice (lanes 1 to 3) and 6-month-old rat EDL and SOL muscles (lanes 4 and 5). (**D**). Forelimb muscles from young (2 months, lanes 4–6) and old (19–20 months, lanes 1–3) WT/WT, WT/MUT and MUT/MUT mice. (**A**–**D**) Mouse muscle homogenates were obtained from 2 mice of the same age and genotype. The mouse homogenate pellets were resuspended at a concentration of 0.6 mg/mL. Rat muscle homogenates are from one rat. The rat homogenate pellets were resuspended at a concentration of 0.3 mg/mL. A 20 µL volume of the resuspended homogenate pellet solution was loaded onto the gels in all cases (**A**–**D**). The position of the 150 kDa and 250 kDa molecular weight markers are shown on the left-hand side of each gel.

**Table 2 ijms-25-00434-t002:** Twitch contractile properties of the fast EDL and slow SOL muscles from WT/MUT and MUT/MUT mice compared heir WT/WT littermates.

Muscle	Genotype	Pt (mN/mm^2^)	TPT (s)	½ RT (s)	Dx/Dt (mN/s)
**EDL**	WT/WT	83.23 ± 9.3	0.018 ± 0.0005	0.015 ± 0.0005	3911 ± 539
	WT/MUT	63.69 ± 9.2	0.017 ± 0.0005	0.015 ± 0.0005	3944 ± 457
	MUT/MUT	61.3 ± 10.7	0.017 ± 0.0004	0.014 ± 0.0005	3952 ± 661
**SOL**	WT/WT	10.4 ± 1.08	0.031 ± 0.003	0.074 ± 0.018	495.6 ± 52.2
	WT/MUT	10.01 ± 0.39	0.033 ± 0.003	0.096 ± 0.017	442.4 ± 27.7
	MUT/MUT	8.96 ± 2.02	0.031 ± 0.001	0.071 ± 0.011	427.2 ± 92.1

Pt, maximum twitch force; TPT, time to peak twitch; ½ RT, one-half relaxation time; Dx/Dt, maximal rate of contraction. Values shown are mean ± standard error of the mean. WT/WT mice (N = 6), WT/MUT mice (N = 4) and MUT/MUT mice (N = 5).

**Table 3 ijms-25-00434-t003:** Summary of fibre types examined in skinned fibre experiments. A summary of fibre types within SOL and EDL muscles from wild type (WT/WT), heterozygous (WT/MUT) and homozygous (MUT/MUT) mice. “n” denotes the number of fibres and “N” the number of animals examined. Values in parentheses are the relative proportion (%) of the total number of fibres within the muscle examined. “#” denotes significant differences using Chi-square (χ^2^) goodness of fit tests (*p* < 0.05). Fibre types were functionally identified by sensitivity to pSr 5.3 (see Section 4 and Figure 3 and Figure 4).

Muscle	Genotype	N	n	Fibre Types			Type I + Type II
		Mice	Fibres	Type II	Type I	Mixed	(% Total Fibres)
**EDL**	WT/WT	4	16	14 (87.5%)	1 (6.3%)	1 (6.3%)	93.8%
	WT/MUT	5	14	12 (85.7%)	2 (14.3%)	0 (0%)	100%
	MUT/MUT	3	12	9 (75%)	3 (25%) #	0 (0%)	100%
**SOL**	WT/WT	4	21	10 (47.6%	9 (42.9%)	2 (9.5%)	90.5%
	WT/MUT	5	15	4 (26.7%)	8 (53.3%)	3 (20%)	80%
	MUT/MUT	3	15	7 (46.7%)	4 (26.7%)	4 (26.7%) #	73.3% #

**Table 4 ijms-25-00434-t004:** Summary of the contractile apparatus properties. Values are mean ± SEM of pCa_50_ and Hill coefficient (*h*) in SOL and EDL fibres from the three cohorts (i.e., WT/WT, WT/MUT and MUT/MUT). “n” denotes the number of fibres and “N” the number of animals examined. “#” indicates a significant increase in Ca^2+^ sensitivity of the contractile apparatus (*p* < 0.05, one-way ANOVA) between WT/WT and MUT/MUT.

Muscle	Genotype	N Mice	n Fibres	pCa_50_	*h*
**EDL**	WT/WT	4	13	5.794 ± 0.017	5.725 ± 0.536
	WT/MUT	5	12	5.849 ± 0.016	5.704 ± 0.491
	MUT/MUT	3	9	5.862 ± 0.009 #	5.872 ± 0.249
**SOL**	WT/WT	4	18	5.789 ± 0.026	3.984 ± 0.224
	WT/MUT	5	12	5.858 ± 0.019	3.660 ± 0.296
	MUT/MUT	3	12	5.868 ± 0.029 #	4.025 ± 0.326

**Table 5 ijms-25-00434-t005:** Parameter values for channels in 1 µM cytoplasmic Ca^2+^, expressed relative to values in 300 nM cytoplasmic Ca^2+^. Data are given as mean ± sem. **#** indicates significant changes in relative parameters at 1 µM cytoplasmic Ca^2+^ compared to control with 300 nM cytoplasmic Ca^2+^ with n listing the number of observations. **Ω** indicates a significant difference from WT/WT. Please refer to the legends of Figure 6, Appendix A for further details. Data are shown for relative *P_o_*, relative (mean open time) (*T_o_*) and relative *F_o_* only as there was no consistent correlation between the changes in relative *P_o_* (Figure 6) and the relative mean closed time (rel *T_c_,*
Appendix A).

	Genotype	Young	Middle-Aged	Old
rel *P_o_*	WT/WT	1.520 ± 0.208 (n = 14) **#**	1.343 ± 0.189 (n = 16) **#**	1.169 ± 0.235 (n = 8)
	WT/MUT	1.990 ± 0.628 (n = 16)	2.073 ± 0.721 (n = 12)	2.049 ± 0.601 (n = 6)
	MUT/MUT	2.066 ± 0.799 (n = 12)	1.649 ± 0.463 (n = 14)	2.792 ± 0.961 (n = 6) **# Ω**
rel *T_o_*	WT/WT	1.111 ± 0.088 (n = 14) **#**	1.068 ± 0.0719 (n = 16)	1.049 ± 0.110 (n = 8)
	WT/MUT	1.251 ± 0.091 (n = 16) **#**	1.142 ± 0.054 (n = 12) **#**	1.250 ± 0.066 (n = 6) **#**
	MUT/MUT	1.032 ± 0.146 (n = 12)	1.032 ± 0.058 (n = 14)	1.648 ± 0.569 (n = 6)
rel *F_o_*	WT/WT	1.399 ± 0.187 (n = 14) **#**	1.405 ± 0.193 (n = 16) **#**	1.043 ± 0.158 (n = 8)
	WT/MUT	1.363 ± 0.312 (n = 16)	1.697 ± 0.576 (n = 12)	1.700 ± 0.619 (n = 6)
	MUT/MUT	2.751 ±1.795 (n = 12)	1.561 ± 0.428 (n = 14)	1.628 ± 0.237 (n = 6) **# Ω**

**Table 6 ijms-25-00434-t006:** Parameter values for channels exposed to 10 uM and 10 mM caffeine expressed relative to values in 300 nM cytoplasmic Ca^2+^, are given as mean ± sem. **#** indicates significant changes relative parameters at 1 µM cytoplasmic Ca^2+^ compared to control at 300 nM cytoplasmic Ca^2+^ with n listing the number of observations. **Ω** indicates significant differences between relative *P_o_* for WT/WT and WT/MUT or MUT/MUT. Please refer to the legends of Figure 6 and Appendix A for further details. Data are shown for relative *P_o_*, relative *T_c_* and relative *F_o_* only as there was no consistent correlation between the changes in relative *P_o_* (Figure 6) and the relative *T_c_*, (Appendix A). Data are shown for 10 µM caffeine to indicate sensitivity and 10 mM to indicate efficacy at 10 mM. The number of observations for each data set are the same as those given in Table 5.

	Genotype	Young		Middle-Aged		Old	
		10 µM caff	10 mM caff	10 µM caff	10 mM caff	10 µM caff	10 mM caff
rel *P_o_*	WT/WT	1.061 ± 0.211	13.010 ± 3.716 **#**	1.480 ± 0.143 **#**	12.652 ± 2.745 **#**	1.215 ± 0.140	6.567 ± 1.541 **#**
	WT/MUT	1.630 ± 0.299 **#**	15.425 ± 3.854 **#**	1.610 ± 0.366 **#**	9.766 ± 3.086 **#**	1.998 ± 0.881	3.848 ± 0.386 **#**
	MUT/MUT	1.164 ± 0.315	13.882 ± 3.643 **#**	1.322 ± 0.194	12.615 ± 3.458 **#**	5.234 ± 1.741 **# Ω**	19.066 ± 5.560 **# Ω**
rel *T_c_*	WT/WT	1.425 ± 0.228	0.209 ± 0.056 **#**	0.964 ± 0.216	0.199 ± 0.038 **#**	0.916 ± 0.081	0.504 ± 0.212 **#**
	WT/MUT	1.035 ± 0.197	0.309 ± 0.087 **#**	0.786 ± 0.071 **#**	0.356 ± 0.099 **#**	1.083 ± 0.327	0.316 ± 0.047 **#**
	MUT/MUT	1.444 ± 0.278	0.166 ± 0.033 **#**	0.942 ± 0.099	0.235 ± 0.035 **#**	0.746 ±0.2696	0.371 ± 0.241 **#**
rel *F_o_*	WT/WT	0.995 ± 0.178	9.026 ± 2.554 **#**	1.374 ± 0.134 **#**	7.061 ± 1.251 **#**	1.170 ± 0.134	6.702 ± 3.105 **#**
	WT/MUT	1.481 ± 0.260 **#**	6.703 ± 1.609 **#**	1.706 ± 0.398 **#**	7.119 ± 2.140 **#**	1.910 ± 0.787	3.657 ± 0.938 **#**
	MUT/MUT	0.885 ± 0.286	9.639 ± 4.412 **#**	1.232 ± 0.136	6.347 ± 0.327 **#**	3.857 ± 1.625	8.155 ± 1.656 **#**

**Table 7 ijms-25-00434-t007:** Cross-sectional areas of fibres in EDL and SOL muscles from 2- to 4-month-old WT/WT, WT/MUT and MUT/MUT mice where n is the number of mice for each genotype. Fibres in each muscle were allocated according to their cross-sectional area to group A (larger) or group B (smaller). In sections of the muscles from each animal, the areas of 34 neighbouring larger group A fibres were measured, and then the interspersed small-diameter group B fibres in the same area of the section were counted and the cross-sectional areas were measured. In EDL muscles, there were 9–35 group B fibres in the same region as the 34 group A fibres. In contrast, in SOL muscles, group B fibres were found in sections from 1 WT/WT mouse and 1 MUT/MUT mouse, so n = 1 mouse, although sections from 14 and 10 mice, respectively, were examined. SOL muscles from 5 of the 15 WT/MUT mice contained group B fibres. *****, group A EDL fibres from WT/MUT or MUT/MUT mice significantly smaller that group A EDL fibres from WT/WT mice. **^@^**, group B EDL fibres significantly smaller than group A EDL fibres. **^#^**, group A SOL fibres significantly smaller than group A EDL fibres.

	EDL Group A µm^2^	EDL Group B µm^2^	SOL Group A µm^2^	SOL Group B µm^2^
WT/WT	808.9 ± 54.4	270.5 ± 28.9 **^@^**	645 ± 24 **^#^**	322
	(n = 14)	(n = 14)	(n = 14)	(n = 1 of 14)
WT/MUT	713.8 ± 56.2 *	270.5 ± 28.8 **^@^**	635 ± 32 **^#^**	354 ± 67
	(n = 15)	(n = 15)	(n = 15)	(n = 3 of 15)
MUT/MUT	670.0 ± 31.4 *	238.5 ± 10.42 **^@^**	566 ± 24 ***^#^**	197
	(n = 10)	(n = 10)	(n = 10)	(n = 1 of 10)

## Data Availability

The data presented in this study is contained within the article and Appendix A.

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
