# Peer review of "The RyR1 P3528S Substitution Alters Mouse Skeletal Muscle Contractile Properties and RyR1 Ion Channel Gating"

_ijms, 2023, doi:10.3390/ijms25010434_

Round 1

Reviewer 1 Report

Comments and Suggestions for Authors

Manuscript "The RyR1 P3528S substitution alters mouse skeletal muscle 2 contractile properties and RyR1 ion channel gating" reports on functional and biochemical characterization of a mouse model a myopathy caused by the recessive mutation P3527S mutation in humans. The mouse model apparently only partially mirrors the human myopathy since, at difference with humans, no evidence of cores or mini-cores are observed in mice. The characterization of the mouse model, however, is suggestive of alterations in muscle strength and fiber composition. The manuscript is potentially relevant, however, some points have to be considered. Unfortunately three of the 10 mentioned figures are missing and have not been evaluated. Figures and results are, sometimes, quite complex; avoiding to comment expected difference (for example due to mouse age and sex, present in wild type mice) may help. Results from strength tests seem interesting and deserve a comment in the text.

Lines 103-104: differences between male and female and between young and old animals are expected. Probably there is no need to underline statistical significance, unless it correlates with the mouse genotype.

When analyzing maximum specific force of EDL and Soleus muscles (line 134), the authors state, in the text, that this was normalized to body and muscle mass, while in the legend to figure 2, they state that it was normalized to muscle cross sectional area. Can they explain if there is a difference between the two approaches?

Line 23: please add "with" after "compared"

line 26: please add "of" before "maximum"; please add "in" before "Ca2+ activation"

line 75: please add "of" before autosomal"

Line 80: please replace "was" with "were"

Results on RyR1 channel activity (paragraph 2.5) are interesting. However, data presentation is quite complex and not easy to follow. We suggest to compare and analyze statistical significance between the three genotypes in a single graph; alternatively, please add a summary table reporting statistical significance between genotypes. Try to emphasize differences among phenotypes rather than in wild type animals.

Figure 7 is missing

Table 5:  Since fiber have been grouped in two size groups, it is expected that differences in size between group A and group B are statistically significant. There is no need to use the green @. Accordingly, fibers from soles and EDL are expected to have a different size. 

Myosin Heavy chain analysis: the use of single muscles (EDL and SOLEUS) would be more informative than total hindlimb muscles. Please replace this analysis with analysis of EDL and Soleus muscles

Figure 8 and 9 are missing

Point 3.4 in discussion: are there evidence of alterations in mutant mouse behavior? if not this section can be deleted

Point 3.7 in discussion: are there evidence of alteration in mutant mouse embryonic development? If not this section can be deleted

Comments on the Quality of English Language

English requires a editing. Please check all sentences.

Author Response

Reviewer #1

We thank the reviewer for their detailed evaluation of the manuscript and for the positive and constructive comments. We have addressed the following comments:

Reviewer comment 1: Unfortunately, three of the 10 mentioned figures are missing.

RESPONSE: Thank you for mentioning this point.  The submitted manuscript contained 8 figures.  Two of the figures (Figure 1 and Figure 7) were left out of the pdf created for Peer review.  There were also errors in reference to figures 9 and 10 in the final sections of the Results. These should have been referred to as figures 7 and 8.  The revised manuscript has been appropriately corrected. 

Reviewer comment 2: Figures and results are, sometimes, quite complex; avoiding commenting on expected difference (for example due to mouse age and sex, present in wild type mice) may help.

RESPONSE: We have attempted to change the text to clarify the results and reduce the emphasis on expected differences, however the expected differences in WT/WT validate the techniques and it is important to note that the expected differences were not changed by the mutation.

Reviewer comment 3. Results from strength tests seem interesting and deserve a comment in the text.

RESPONSE: Results from strength tests are addressed in the Abstract, Results and Discussion under appropriate headings.

Reviewer comment 4. Lines 103-104: differences between male and female and between young and old animals are expected. Probably there is no need to underline statistical significance, unless it correlates with the mouse genotype.

RESPONSE: “significantly” has been removed from the text.

Reviewer comment 5.  When analyzing maximum specific force of EDL and Soleus muscles (line 134), the authors state, in the text, that this was normalized to body and muscle mass, while in the legend to figure 2, they state that it was normalized to muscle cross sectional area. Can they explain if there is a difference between the two approaches?

RESPONSE:  The text has been corrected and now reads “At frequencies above 60Hz in EDL, and above 30Hz in the SOL, there was no difference in force production. When normalised to muscle cross sectional area, maximum specific force (sFmax) of WT/MUT and MUT/MUT EDL muscles was significantly lower than WT/WT EDL muscles (P<0.05, Fig 2C). 

Reviewer comment 6. Line 23: please add "with" after "compared".

RESPONSE:  Thank you, but this comment is no longer relevant because of changes in the abstract in response to Reviewer #2.

Reviewer comment 7. line 26: please add "of" before "maximum"; please add "in" before "Ca2+ activation".

RESPONSE:  Thank you, but again the abstract has been changed in response to Reviewer #2.

Reviewer comment 8.  line 75: please add "of" before autosomal

RESPONSE:  "of" has been added before autosoma.l

Reviewer comment 9. Line 80: please replace "was" with "were".

RESPONSE: This section is now removed to accommodate the request from Reviewer #2.   

Reviewer comment 10.  Results on RyR1 channel activity (paragraph 2.5) are interesting. However, data presentation is quite complex and not easy to follow. We suggest comparing and analyzing statistical significance between the three genotypes in a single graph; alternatively, please add a summary table reporting statistical significance between genotypes. Try to emphasize differences among phenotypes rather than in wild type animals.

RESPONSE:  Tables 5 and 6 summarising statistical significance between genotypes has been added.  And differences between phenotypes have been emphasised in the text.  We have retained the mention of age-related differences in WT/WT this is an interesting observation and worth emphasising.  

Reviewer comment 11. Figure 7 is missing.

RESPONSE:  Please see response to point 1.

Reviewer comment 12. Table 5:  Since fiber have been grouped in two size groups, it is expected that differences in size between group A and group B are statistically significant. There is no need to use the green @. Accordingly, fibers from soles and EDL are expected to have a different size.

RESPONSE:  NB this is now Table 7. The division of fibres into two size groups reported here has not been described previously and is thus not expected.  The significant difference between sizes justifies the division into the two groups.  Thus, the significance is retained.  It is useful to also confirm the expected differences between EDL and soleus to confirm that the results from our study and that our particular analysis fits with previous studies and threrfore justify the conclusions that there are differences in genotypeps.

Reviewer comment 13.  Myosin Heavy chain analysis: the use of single muscles (EDL and SOLEUS) would be more informative than total hindlimb muscles. Please replace this analysis with analysis of EDL and Soleus muscles.

RESPONSE:  The data from total limb muscle analysis is important as it suggests that the mutation has specific effects on particular muscle groups rather than a global change in fiber types.  We agree that the data from EDL and soleus would be informative and mention this in both the Results and in the possible limiting factors section now included at the end of the Discussion.

Reviewer comment 14.  Figures 8 and 9 are missing.

RESPONSE:  Please see response to point 1. 

Reviewer comment 14.  Point 3.4 in discussion: are there evidence of alterations in mutant mouse behavior? if not this section can be deleted.

RESPONSE:  Yes, as mentioned in the Results and the Discussion, the results of the strength tests alone could reflect a behavioural change.  The section has been retained.

Reviewer comment 15.  Point 3.7 in discussion: are there evidence of alteration in mutant mouse embryonic development? If not, this section can be deleted.

RESPONSE:  There is no evidence for embryonic development in the P3528S mouse.  However we would like to retain the section as other mice that we attempted to generate did show such signs.  Therefore we would like to retain this small segment as it might be useful to others attempting to generate models of other RyR1 mutations.

Reviewer comment 16.  English requires a editing. Please check all sentences.

RESPONSE:  The English has been thoroughly checked and typos corrected. All errors mentioned were the result of typos rather than poor English.  Please note that we have replaced “fiber” with “fibre” throughout as it is that is the accepted English spelling in most countries other than the US.  We were unable to find any specific recommendation from the journal in that case.  

Reviewer 2 Report

Comments and Suggestions for Authors

General Comments

This is an interesting study, evaluating the effects of the mutation in an equivalent RyR1 P3528S in mouse on mobility and strength, isolated muscle contractile and histological profiles, fiber-types, myosin isoforms and RyR1 channel activity. CRISPR-Cas9 gene editing was used to generate the P3528S mouse. The wild type (WT/WT, n=), heterozygous (WT/MUT) and homozygous (MUT/MUT) mice were examined. When WT/MUT and MUT/MUT characteristics were compared wild type (WT/WT), there were: (1) a reduction in strength without changes in physical characteristics or longevity;  (2) a shift in force frequency responses of extensor digitorum longus (EDL) and soleus (SOL) muscles to higher frequencies; (3) a reduced maximum specific force in the EDL; (4) a shift Ca2+ activation of skinned EDL and SOL fibers to lower Ca2+; (5) increased percentages of type I fibers in EDL and mixed type I/II fibers in SOL with associated histological changes, but no change in total myosin heavy chain composition; (6) a shift in the threshold for Ca2+ activation of RyR1 channels to lower Ca2+; (7) a loss of age-dependent increases channel activity. The Authors concluded that a significant changes were observed in all parameters in both heterozygous and homozygous situations, indicating that the presence of the substitution on only one or two of the four subunits of the RyR1 protein was sufficient to impact on the gating of the ion channel with flow-on effects on Ca2+ homeostasis and from there to a variety of other muscle characteristics.

This study contains new data in this context, and the manuscript is generally well written. However, the design of this paper should be little improved before publishing.

1.    The Authors should clearly present the main aim of and novelty (hypotheses ?) of the present study in the end of the Introduction (Page 2, Lines 78-96) rather than only generally describe the animals and methods used. I suggest to decrease this descriptive part.

2.    In my opinion, it is obligatory to add limiting factors of the study at the end of the Discussion chapter. The most important of them seems to be the heterogeneity of the number of animals in measured groups of mice (from 3 for MUT/MUT young male to 15 for WT/WT old male) and very small sample size for some measured groups (n=3 for MUT/MUT young male). These factors can influence the results of the present study and, therefore, it is obligatory to note and analyze in Discussion.

Specific Comments

Abstract

Page 1. Please add the number of mice for WT/WT, WT/MUT and MUT/MUT groups.

1.Introduction

Page 2, Lines 78-96. Please present the main aim and novelty (hypotheses?) of this study in the end of the Introduction (see General Comments)

2. Results

Page 3. Table 1. Please add the units of measurement of the average weight of mice for WT/WT, WT/MUT and MUT/MUT groups.

 3.Discussion

Page 21. Please add possible limiting factors of this study at the end of the Discussion chapter (see General Comments).

Author Response

Reviewer #2

We thank the reviewer for their evaluation of the manuscript the positive and for the constructive responses and comments. We have addressed the comments as follows:

Reviewer comment 1.  The Authors should clearly present the main aim of and novelty (hypotheses?) of the present study in the end of the Introduction (Page 2, Lines 78-96) rather than only generally describe the animals and methods used. I suggest decreasing this descriptive part.

RESPONSE:  The final paragraph of the Introduction has been rewritten as requested.

Reviewer comment 2.  In my opinion, it is obligatory to add limiting factors of the study at the end of the Discussion chapter. The most important of them seems to be the heterogeneity of the number of animals in measured groups of mice (from 3 for MUT/MUT young male to 15 for WT/WT old male) and very small sample size for some measured groups (n=3 for MUT/MUT young male). These factors can influence the results of the present study and, therefore, it is obligatory to note and analyze in Discussion.

RESPONSE:  A ‘limiting factors’ section has been added to the Discussion with emphasis on the heterogeneity of the number of animals.

Reviewer comment 3.  Abstract, Page 1. Please add the number of mice for WT/WT, WT/MUT and MUT/MUT groups.

RESPONSE:  Overall mouse numbers have been added to the abstract and the abstract substantially rewritten to accommodate this change within the ~200-word limit.   The overall numbers are used because the specific numbers varied from experiment to experiment depending on availability of mice and the demands of the experiment.  The specific numbers are given throughout the manuscript as appropriate in Figure legends and Tables.  The variability in numbers between experiments is also addressed in the “limiting factors” section at the end of the Discussion as requested.

Reviewer comment 4.  Introduction. Page 2, Lines 78-96. Please present the main aim and novelty (hypotheses?) of this study in the end of the Introduction (see General Comments).

RESPONSE:  Please see response to point 1 above.

Reviewer comment 5.  Results Page 3. Table 1. Please add the units of measurement of the average weight of mice for WT/WT, WT/MUT and MUT/MUT groups.

RESPONSE:  Units of measurement added to average weight in Table 1.

Reviewer comment 6.  Discussion. Please add possible limiting factors of this study at the end of the Discussion chapter (see General Comments).

RESPONSE:  Please see response to point 2 above.  A “possible limiting factors section” has been added at the end of the Discussion.